# Input Invex Neural Network

## Abstract

Connected decision boundaries are useful in several tasks like image segmentation, clustering, alpha-shape or defining a region in nD-space. However, the machine learning literature lacks methods for generating connected decision boundaries using neural networks. Thresholding an invex function, a generalization of a convex function, generates such decision boundaries. This paper presents two methods for constructing invex functions using neural networks. The first approach is based on constraining a neural network with Gradient Clipped-Gradient Penality (GCGP), where we clip and penalise the gradients. In contrast, the second one is based on the relationship of the invex function to the composition of invertible and convex functions. We employ connectedness as a basic interpretation method and create connected region-based classifiers. We show that multiple connected set based classifiers can approximate any classification function. In the experiments section, we use our methods for classification tasks using an ensemble of 1-vs-all models as well as using a single multiclass model on larger-scale datasets. The experiments show that connected set-based classifiers do not pose any disadvantage over ordinary neural network classifiers, but rather, enhance their interpretability. We also did an extensive study on the properties of invex function and connected sets for interpretability and network morphism with experiments on simulated and real-world data sets. Our study suggests that invex function is fundamental to understanding and applying locality and connectedness of input space which is useful for various downstream tasks.

## 1 Introduction

The connected decision boundary is one of the simplest and most general concepts used in many areas, such as image and point-cloud segmentation, clustering or defining reason in nD-space. Connected decision boundaries are fundamental to many concepts, such as convex-hull and alpha-shape, connected manifolds and locality in general. Yet another advantage of the connected decision boundaries is selecting a region in space. If we want to modify a part of a function without changing a global space, we require a simply connected set to define a local region. Such local regions allow us to learn a function in an iterative and globally independent way. Similarly, locality-defining neurons in Neural Networks are useful for Network Morphism based Neural Architecture Search (Wei et al., 2016; Chen et al., 2015; Cai et al., 2018; Jin et al., 2019; Elsken et al., 2017; Evci et al., 2022; Sapkota & Bhattarai, 2022). These characteristics of decision boundaries can be crucial for the interpretability and explainability of neural networks. It is evident that Neural Networks are de-facto tools for learning from data. However, the concerns over the model being black-box are also equally growing. Furthermore, machine learning models such as clustering and classification are based on the smoothness assumption that the local space around a point also belongs to the same class or region. However, such regions are generally defined using simple distance metrics such as Euclidean distance, which limits the capacity of the models using it. In reality, the decision boundaries can be arbitrarily non-linear, as we observe in image segmentation, alpha-shape and too few to mention.

Let us introduce connected set, disconnected set and convex set as decision boundaries by comparing them side by side in Figure 1. From the Figure, we can observe that the convex set is a special case of the connected set and the union of connected sets can represent any disconnected set. And, the disconnected sets are sufficient for any classification and clustering task, yet the individual sets are still connected and

interpretable as a single set or region. Furthermore, we can relate the concept of *simply connected* (1-connected) decision boundary with the locality and also be viewed as a discrete form of the locality. This gives us insights into the possibilities of constructing arbitrarily non-linear decision boundaries mathematically or programmatically

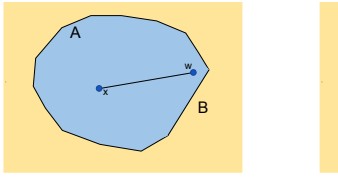 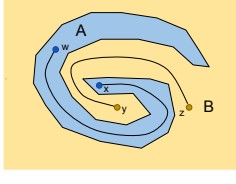 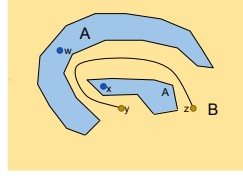 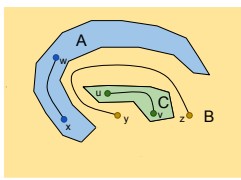

|  (a) Convex Set | (b) Connected Set | (c) Disconnected Set | (d) Connected Sets |

Figure 1: Different types of sets according to the decision boundary in continuous space. **(a)** Convex sets have all the points inside a convex decision boundary. A straight line connecting any two points in the set also lies inside the set. Here, A is a convex set, and B is a non-convex set. **(b)** Connected sets have continuous space between any two points within the set. Any two points in the connected set can be connected by a curve that also lies inside the set. Here, both A and B are connected sets; A is a bounded 1-connected set. **(c)** Disconnected sets are opposite of connected sets. Any two points in the disconnected set can not be connected by a curve that also lies inside the set. Here, A is the disconnected set and B is the connected set. **(d)** The same decision boundary as (c) is represented by multiple connected sets. Disconnected set A in (c) is a union of connected set A and C in (d). Here, all A, B and C are connected sets. However, $A \cup C$ is a disconnected set and $B \cup C$ is still a connected set.

To create a connected decision boundary for low and high-dimensional data, we require a method to constrain neural networks to produce simply connected (or 1-connected) decision boundaries. Our search for a simply connected decision boundary is inspired by a special property of a convex function exhibiting its lower contour set forming a convex set, which is also a simply connected space. However, we need a method that gives us a convex set that is always *simply connected* but not necessarily required to be a convex set.

Invex function (Hanson, 1981; Ben-Israel & Mond, 1986; Mishra & Giorgi, 2008) is a generalization of the convex function that exactly satisfies our above-mentioned requirements. The lower contour set of an invex function is simply connected and can be highly non-linear, unlike convex sets. Simply connected set is equivalent to invex set (Mititelu, 2007; Mohan & Neogy, 1995). However, there does not exist a principled method to constrain a neural network to be invex. To this end, we propose two different methods to constrain the neural network to be invex with respect to the inputs which we call Input Invex Neural Networks (II-NN): (i) Using Gradient Clipped Gradient Penalty (GC-GP) (ii) Composing Invertible and Convex Functions. Among these two methods, GC-GP constrains the gradient of a function with reference to a simple invex function to avoid creating new global optima. This method limits the application to a single variable output and one-vs-all classification. Similarly, our second approach leverages the invertible and convex functions to approximate the invex function. This method is capable of creating multiple simply connected regions for multi-class classification tasks.

We performed extensive both qualitative and quantitative experiments to validate our ideas. For quantitative evaluations, we applied our method to both the synthetic and challenging real-world image classification data sets: MNIST, Fashion MNIST, CIFAR-10 and CIFAR-100. We compared our performance with competitive baselines: Ordinary Neural Networks and Convex Neural Networks. Although we validated our method for classification tasks, our method can be extended to other tasks too. This is well supported by both the parallel works (Nesterov et al., 2022; Brown et al., 2022) and earlier work (Izmailov et al., 2020) using the invex function and simply connected sets as a fundamental concept for tasks other than classification. Similar to our study of simply connected decision boundaries, simply connected manifolds are homeomorphic to the n-Sphere manifold and are related to the Poincaré conjecture (See Appendix Sec. I). We design our qualitative experiments to support our claims on the interpretability and explainability of our method compared to our Neural baseline architectures. Please note comparing our method with the work on AI and explainability would be beyond the scope.

Overall, we summarize the contributions of this paper as follows:

1. We introduced the invex function in the Neural Networks. To the best of our knowledge, this is the first work introducing the invex function on Neural Networks.
2. We present two methods to construct invex functions, compare their properties along with convex and ordinary neural networks as well as demonstrate its application for interpretability.
3. We present a new type of supervised classifier called multi-invex classifiers which can classify input space using multiple simply connected sets as well as show its application for network morphism.
4. We experiment with classification tasks using toy datasets, MNIST, FMNIST, CIFAR-10 and CIFAR-100 datasets and compare our methods to ordinary and convex neural networks.

## 2 Background and Related Works

### 2.1 Locality and Connected Sets

This work on the invex function is highly motivated by the connected set and its applications. A better mathematical topic related to what we are looking for is the simply connected set or space. This type of set is formed without any holes in the input space inside the defined set. Some examples of simply connected space are convex space, euclidean plane or n-sphere. Other space such as the torus is not simply connected space, however, they are still connected space.

The concept of locality is another idea related to simply connected space. In many machine learning algorithms, the locality is generally defined by euclidean distance. This gives the nearest examples around the given input $x$ or equivalently inside some connected n-Sphere. This concept of locality is used widely in machine learning algorithms such as K-NN, K-Means, RBF (Broomhead & Lowe, 1988). The concept of locality has been applied in local learning (Bottou & Vapnik, 1992; Cleveland & Devlin, 1988; Ruppert & Wand, 1994). These works highlight the benefit of local learning from global learning in the context of Machine Learning. The idea is to learn local models only from neighbourhood data points. Although these methods use simple methods of defining locality, we can define it arbitrarily.

We try to view these two concepts, locality and simply connected space, as related and interdependent on some locality-defining function. In traditional settings, the locality is generally defined by thresholding some metric function, which produces a bounded convex set. In our case, we want to define locality by any bounded simply connected space which is far more non-linear. Diving into metric spaces is beyond the topic of our interest. However, we connect both of these ideas with the generalized convex function called the invex function. We use a discrete form of locality, a connected region, produced by the invex function to classify the data points inside the region in a binary connected classifier and multiple connected classifiers in Section 3.

### 2.2 Generalized Convex Functions

We start this subsection with the general definition of Convex, Quasi-Convex and Invex functions.

**Convex Function**   *A function on vector space $\mathbf{X} \in \mathbb{R}^n$, $f : \mathbf{X} \to \mathbb{R}$ is convex if:*
*For all $\mathbf{x_1}, \mathbf{x_2} \in \mathbf{X}$, and $t \in [0, 1]$,*

$$f(t\mathbf{x_1} + (1 - t)\mathbf{x_2}) \leq tf(\mathbf{x_1}) + (1 - t)f(\mathbf{x_2}) \tag{1}$$

**Quasi-Convex Function**   *A function on vector space $\mathbf{X} \in \mathbb{R}^n$, $f : \mathbf{X} \to \mathbb{R}$ is quasi-convex if:*
*For all $\mathbf{x_1}, \mathbf{x_2} \in \mathbf{X}$, and $t \in [0, 1]$,*

$$f(t\mathbf{x_1} + (1 - t)\mathbf{x_2}) \leq \max\{f(\mathbf{x_1}), f(\mathbf{x_2})\} \tag{2}$$

If we replace the inequality with '$<$' then, the function is called **Strictly Quasi-Convex Function**. Such a function does not have a plateau, unlike the quasi-convex function.

**Invex Function**     *A differentiable function on vector space* $\mathbf{X} \in \mathbb{R}^n$, $f : \mathbf{X} \to \mathbb{R}$ *is invex if:*

*For all* $\mathbf{x_1}, \mathbf{x_2} \in \mathbf{X}$, *and there exists an invexity function* $\eta : \mathbf{X} \times \mathbf{X} \to \mathbb{R}^n$,

$$f(\mathbf{x_1}) - f(\mathbf{x_2}) \geq \eta(\mathbf{x_1}, \mathbf{x_2}) \cdot \nabla f(\mathbf{x_2}) \tag{3}$$

Invex function with invexity $\eta(\mathbf{x_1}, \mathbf{x_2}) = \mathbf{x_1} - \mathbf{x_2}$ is a convex function (Ben-Israel & Mond, 1986). Following are the 2 properties of the invex function that are relevant for our experiments. Furthermore, non-differentiable functions also invex.

**Property 1** *If* $f : \mathbb{R}^n \to \mathbb{R}^m$ *with* $n \geq m$ *is differentiable with Jacobian of rank* $m$ *for all points and* $g : \mathbb{R}^m \to \mathbb{R}$ *is invex, then* $h = g \circ f$ *is also invex.*

This property can be simplified by taking $m = n$, which makes the function $f$ an invertible function. Furthermore, the function with Jacobian of rank $m$ is simply learning the $m$-dimensional manifold in the $n$-dimensional space (Brehmer & Cranmer, 2020). The above property simplifies as follows .

*If* $f : \mathbb{R}^n \to \mathbb{R}^n$ *is differentiable, invertible and* $g : \mathbb{R}^n \to \mathbb{R}$ *is invex, then* $h = g \circ f$ *is also invex.*

**Property 2** *If* $f : \mathbb{R}^n \to \mathbb{R}$ *is differentiable, invex and* $g : \mathbb{R} \to \mathbb{R}$ *is always-increasing, then* $h = g \circ f$ *is also invex.*

However, during experiments, we use a non-differentiable continuous function like ReLU which has continuous counterparts. Such functions do not affect the theoretical aspects of previous statements.

The invex function and quasi-convex function are a generalization of a convex function. All convex and strongly quasi-convex functions are invex functions, but the converse is not true. These functions inherit a few common characteristics of the convex function as we summarize in Table 1. We can see that all convex, quasi-convex and invex functions have only global minima. Although the quasi-convex function does not have an increasing first derivative, it still has a convex set as its lower contour set.

Invex sets are a generalization of convex sets. However, we take invex sets as the lower contour set of the invex function, which for our understanding is equivalent to a simply connected set. There has been a criticism of the unclear definition of invexity and its related topics such as pre-invex, quasi-invex functions and sets (Zălinescu, 2014). However, in this paper, we are only concerned with the invex function and invex or simply connected sets. We define the invex function as the class of functions that have only global minima as the stationary point or region. The minimum/minima can be a single point or a simply connected space.

As mentioned before, the lower contour set of the invex function is a simply connected set, which is a general type of connected decision boundary. Such a decision boundary is more interpretable as it has only one set and can form a more complex disconnected set by a union of multiple connected sets. We present an example of invex, quasi-convex and ordinary functions in Figure 2. Their class decision boundary/sets are compared in Table 1.

Table 1: Comparison of property of Convex, Quasi-Convex, Invex and Ordinary Function.

| Function Type | Only Global Minima | Increasing First Derivative | Set $(-\infty, \theta)$ |
|---|---|---|---|
| Convex | ✓ | ✓ | convex |
| Quasi-Convex | ✓ | ✗ | convex |
| Invex | ✓ | ✗ | 1-connected (invex) |
| Ordinary function | ✗ | ✗ | disconnected |

### 2.3   Constraints in Neural Networks

Constraining neural networks has been a key part of the success of deep learning. It has been well studied that Neural Networks can approximate any function (Cybenko, 1989), however, it is difficult to constrain Neural Networks to get desirable properties. It can be constrained for producing special properties such as Convex or Invertible, which has made various architectures such as Invertible Residual Networks (Behrmann

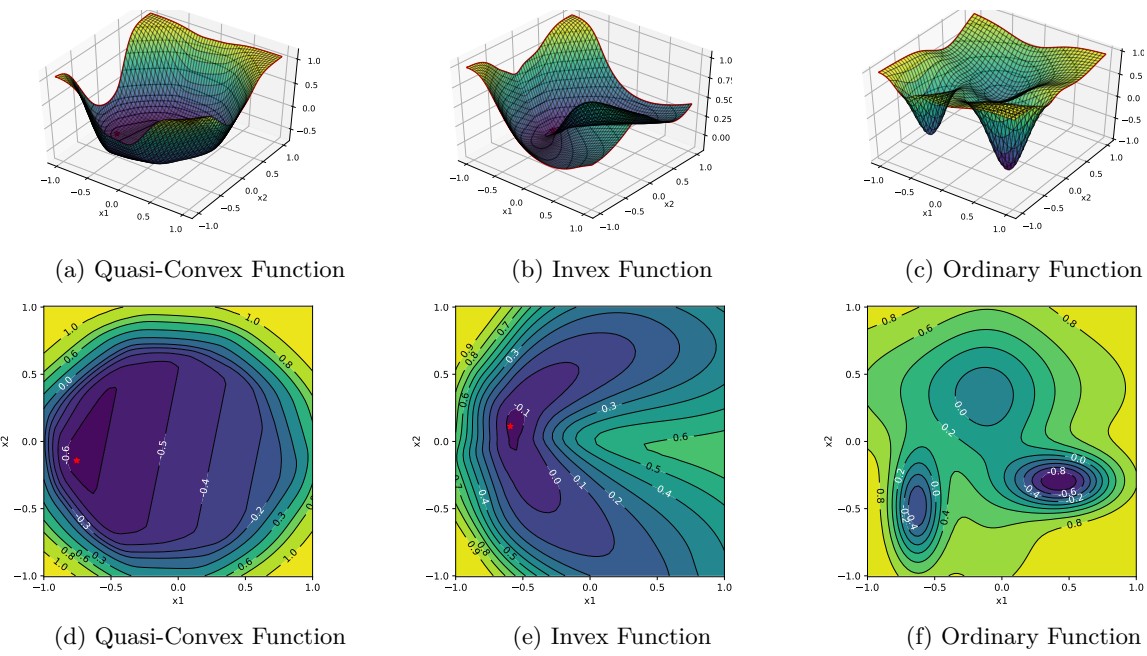

(a) Quasi-Convex Function       (b) Invex Function       (c) Ordinary Function

(d) Quasi-Convex Function       (e) Invex Function       (f) Ordinary Function

Figure 2: 3D plot (top row) and Contour plot (bottom row) of Quasi-Convex, Invex and Ordinary Function. The global minima (red star) is plotted in Convex and Invex Functions. Contour plots on different levels show the decision boundary made by each class of functions. *Zoom in the diagram for details.*

et al., 2019), Convex Neural Networks (Amos et al., 2017), Convex Potential Flows (Huang et al., 2020) possible. Similarly, other areas such as optimization and generalization are made possible by constraints such as LayerNorm (Ba et al., 2016), BatchNorm (Ioffe & Szegedy, 2015) and Dropout (Srivastava et al., 2014).

The Lipschitz constraint of Neural Networks is another important constraint that has received much attention after one of the seminal works on Wasserstein Generative Adversarial Network (WGAN) (Arjovsky et al., 2017). The major works on constraining the Lipschitz constant on neural networks include WGAN-GP (Gulrajani et al., 2017), WGAN-LP (Petzka et al., 2018) and Spectral Normalization (SN) (Miyato et al., 2018). These methods have their benefits and drawbacks. LP is an improvement over GP, where the gradient magnitude is constrained to be below specified K using gradient descent. However, these methods can not constrain the gradients exactly as it is applied with an additional loss function. These methods can however be modified to constrain gradients locally. Furthermore, SN constrains the upper bound of the Lipschitz Constant (gradient norm) globally. This is useful for multiple applications such as on WGAN and iRes-Net (Behrmann et al., 2019). However, our alternative method of constructing the invex function requires constraining the gradient norm locally. Although we can not constrain the local gradients exactly, we refine the gradient constraint to solve the major drawbacks of GP and LP. The details regarding the problems of GP, LP and SN and how our method GC-GP solves the problem are included in the Appendix A.

### 2.4 Convex and Invertible Neural Netowrks

**Convex Neural Network:** ICNN (Amos et al., 2017) has been a goto method for constructing convex functions using neural networks. It has been used in multiple applications such as Convex Potential Flow, Partial Convex Neural Network and Optimization. It is also useful for the construction of invex neural networks when combined with invertible neural networks as mentioned in the previous section.

**Normalizing Flows and Invertible Neural Networks:** Normalizing Flows are one of the bidirectional probabilistic models used for generative and discriminative tasks. They have been widely used to estimate the probability density of training samples, to sample and generate from the distribution and to perform a

discriminative task such as classification. Normalizing Flows use Invertible Neural Networks to construct the flows. It requires computing the log determinant of Jacobian to estimate the probability distribution.

In our case, we only require the neural network to be invertible and do not require computing the log determinant of the Jacobian. This property makes training Invex Neural Networks much more practical, even for large datasets. We can easily compose an Invertible Neural Network and Convex Neural Network to produce Invex Neural Network. This property is supported by the Property 1 of the invex function.

There are many invertible neural network models such as Coupling Layer (Dinh et al., 2014), iRevNet (Jacobsen et al., 2018) and Invertible Residual Networks(iResNet) (Behrmann et al., 2019). iResNet is a great choice for application to the invertible neural network. Since it does not require computing the jacobian, it is relatively efficient to train as compared to training normalizing flows. Furthermore, we can simply compose iResNet and a convex cone function to create an invex function.

**GMM Flow and interpretability:** Gaussian Mixture Model on top of Invertible Neural Network has been used for normalizing flow based semi-supervised learning (Izmailov et al., 2020). It uses a gaussian function on top of an invertible function (an invex function) and is trained using Normalizing Flows. The method is interpretable due to the clustering property of standard gaussian and low-density regions in between clusters. We also find that the interpretability is because of the simply connected decision boundary of the gaussian function with identity covariance. If the covariance is to be learned, the overall functions may not have multiple simply connected clusters and hence are less interpretable. The conditions required for connected decision boundaries are discussed more in Appendix E.

The authors do not connect their approach with invexity, connected decision boundaries or with the local decision boundary of the classifier. However, we alternatively propose that the interpretability is also due to the nature of the connected decision boundary produced by their approach.

Furthermore, the number of clusters is equal to the number of Gaussians and is exactly known. As compared to neural networks, where the decision boundaries are unknown and hence uninterpretable. The simple knowledge that there are exactly N local regions each with their class assignment is the interpretability. We also find that this interpretable nature is useful for analyzing the properties of neural networks such as the local influence of neurons as well as neuron interpretation, which is simply not possible considering the black-box nature of ordinary neural network classifiers.

## 2.5 Classification Approaches

One-vs-All classification (Rifkin & Klautau, 2004) approach is generally used with Binary Classification Algorithms such as Logistic Regression and SVM (Cortes & Vapnik, 1995). However, in recent Neural Network literature, multi-class classification with linear softmax classifier on feature space is generally used for classification. In one of our methods in Section 4.2, we use such One-vs-All classifiers using invex, convex and ordinary neural networks.

Furthermore, we extend the idea of region/leaf-node based classification to simply connected space. This method is widely used in decision trees and soft decision trees (Frosst & Hinton, 2017; Kontschieder et al., 2015). The goal is to assign a probability distribution of classes to each leaf node where each node represents a local region on input space. In our multi-invex classifier, we employ multiple leaf nodes which are then used to calculate the output class probability. This method allows us to create multiple simply connected sets on input space each with its own class probabilities. The detail of this method is in Section 3.2.2. Figure 3 compares the classification approach used generally in Neural Networks to Region based classification.

## 3 Methodology

In this section, we present our two approaches to constructing the invex function. Furthermore, we use the invex function to create a simply connected set for binary classification and multi-class classification models.

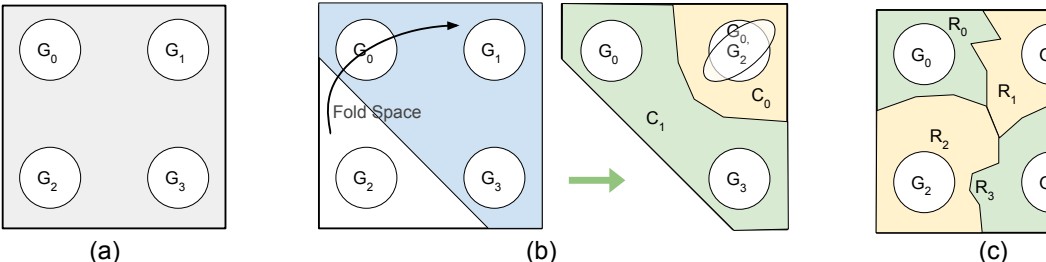

Figure 3: Classification of group of data points ($G_i$) in *(a)* by Ordinary Neural Networks in *(b)* and by Region-based classification method in *(c)*. Here, **(a)** shows the input space of XOR type toy dataset where $G_0$ and $G_3$ belong to class $C_1$ and the rest to class $C_0$. **(b)** Shows two-stage classification by Ordinary Neural Networks, where the input space is transformed (folded) to a two-class cluster and separated by simple classifiers. **(c)** Shows a different approach based on Region based classification. Each of the regions (connected sets) $R_i$ is assigned a class label. This approach still uses neural networks for non-linear morphing of the input space, however, it does not fold the space but only morphs it; the disconnected sets of the same class should be assigned to different regions.

### 3.1 Constructing Invex Function

We present two methods for constructing the invex function. These methods are realised using neural networks and can be used to construct invex functions for an arbitrary number of dimensions.

#### 3.1.1 Invex Function using GC-GP

We propose a method to construct an invex function by modifying an existing invex (or convex) function using its gradient to limit the newly formed function. This method was realised first while constructing invex neural networks as compared to our next method. It consists of 3 propositions to create an arbitrarily complex invex function and does not require an invertible neural network or even a convex neural network. The proposition for constructing the invex function using this method is as follows.

**Proposition 1** *Let $f : \mathbf{X} \to \mathbb{R}$ and $g : \mathbf{X} \to \mathbb{R}$ be two functions on vector space $\mathbf{X}$. Let $\mathbf{x} \in \mathbf{X}$ be any point, $\mathbf{x}^*$ be the minima of $f$ and $\mathbf{x} \neq \mathbf{x}^*$. If $f$ be an invex function and If*

$$\left( \frac{\nabla g(\mathbf{x}) \cdot \nabla f(\mathbf{x})}{\|\nabla f(\mathbf{x})\|} \right) + \|\nabla f(\mathbf{x})\| > 0$$

*then $h(\mathbf{x}) = f(\mathbf{x}) + g(\mathbf{x})$ is an invex function*

**Proposition 2** *Let $g : \mathbf{X} \to \mathbb{R}$ be a function on vector space $\mathbf{X}$, let $\mathbf{x} \in \mathbf{X}$ be any point, $\mathbf{x}^*$ be the minima of $g$ and $\mathbf{x} \neq \mathbf{x}^*$. If*

$$\nabla g(\mathbf{x}) \cdot \frac{\mathbf{x} - \mathbf{x}^*}{\|\mathbf{x} - \mathbf{x}^*\|} > 0$$

*then $g$ is an invex function.*

**Proposition 3** *Modifying invex function as shown in **proposition 1** for N iterations, starting with non-linear $g_i(\mathbf{x})$ and after each iteration, setting $f_{i+1}(x) = h_i(x)$ can approximate any invex function.*

The minima $\mathbf{x}^*$ used in Proposition 2 is the centroid of the invex function in input space which can be easily visualized. The motivations, intuitions and proof for these Propositions are in Appendix C.

**Gradient Clipped Gradient Penalty (GC-GP):** This is our idea to constrain the input gradient value that is later used to construct invex neural network using the above Propositions 1, 2, 3.
*Gradient Penalty:* To penalize the points which violate the projected gradient constraint, we modify the Lipschitz penalty (Petzka et al., 2018). We create a smooth penalty based on the function shown in Figure 4

and Equation 4. We use the smooth-l1 loss function on top of it for the penalty. This helps the optimizer to have a smooth gradient and helps in optimization. It is modified to regularize the projected gradient as well in our II-NN.

$$f_{out\_clip}(pg) = \begin{cases} \dfrac{1}{20} f_{softplus}(20 \cdot pg) & \text{if } pg < 0.14845 \\ 3 \cdot pg - 0.0844560006 & \text{otherwise} \end{cases} \tag{4}$$

*Gradient Clipping:* To make the training stable and not have a gradient opposing the projected gradient constraint, we construct a smooth gradient clipping function using softplus (Dugas et al., 2001) function for smooth transition of the clipping value as shown in Figure 4 and Equation 5. The clipping is done at the output layer before back-propagating the gradients. We clip the gradient of the function to near zero when the K-Lipschitz or projected-gradient constraint is being violated at that point. This helps to avoid criterion gradients opposing our constraint.

$$f_{pg\_penalty}(pg) = \frac{-1}{4} f_{softplus}(-20 \cdot (pg - 0.1)) \tag{5}$$

Combining these two methods allows us to achieve very accurate gradient constraints on the neural network. Our method can not guarantee that the learned function has desired gradient property such as local K-Lipschitz or projected-gradient as defined. However, it constrains the desired property with near-perfect accuracy. This is shown experimentally in Table 5. Still, we can easily verify the constraint at any given point. The Figure 4 shows the step-by-step process for constructing Basic Invex function using Proposition 2 realised using Algorithm 1. The algorithms for constructing different invex functions using this method are in Appendix C.3 and more details of GC-GP are in Section A.

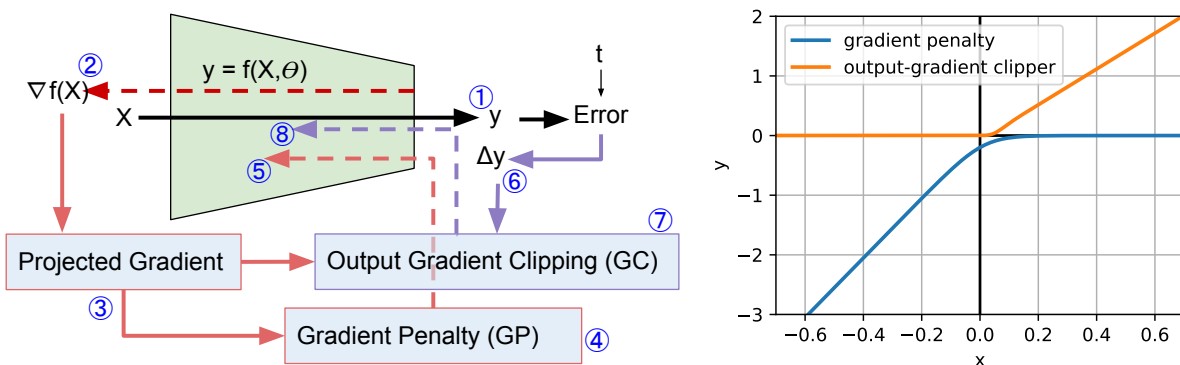

Figure 4: **Left**: Pipeline for Basic II-NN with corresponding pseudo-code in Algorithm 1. **Right**: Function used for output gradient-clipping (GC) and projected gradient-penalty (GP) where, x-axis is the projected-gradient value.

### 3.1.2 Invex Function using Invertible Neural Networks

Invex functions can be created by composing the Invertible function and convex function as discussed in Section 2.4. The choice of invertible function and convex function can both be neural networks, i.e. invertible neural network and convex neural network. We have iResNet and ICNN for invertible and convex neural networks respectively. Composing these two neural networks allows us to learn any invex function.

Furthermore, We can use a simple convex cone with an invertible neural network to create an invex function. In this paper, we use the convex cone as it is relatively simple to implement as well as allows us to visualize the center of the cone in the input space, which represents a central concept for classification and serves as a method of interpretation.

$$f_{invex}(X) = f_{cone}(f_{invertible}(X))$$

**Algorithm 1:** Basic II-NN (Proposition 2) - PyTorch like pseudocode

```
# X, t is the dataset with m elements.
# f_iinn is a parametric neural network model with additional center parameter.
# lamda is the scaling parameter for projected gradient penalty.
# f_pg_scale and f_out_clip are gradient penalty and
#  output_gradient clipper respectively as shown in Figure 4
for step in range(STEPS):
    y = f_iinn(X)    #1
    grad_X = torch.autograd.grad(y, X)   #2
    grad_center = (X - center)
    grad_center = grad_center/torch.norm(grad_center, dim=1)
    pg = torch.batch_dot_product(grad_X, grad_center)    #3
    pgp = f_smoothl1(f_pg_scale(pg)).mean() * lamda   #4
    pgp.backward_to(f_iinn.parameters())     #5
    del_y = criterion(y, t).backward_to(y)    #6
    clip_val = f_out_clip(pg)
    del_y_clip = del_y.clip(-clip_val, clip_val)    #7
    del_y_clip.backward_to(f_iinn.parameters())   #8
    optimizer.step()
```

### 3.2 Simply Connected Sets for Classification

The main goal of this paper is to use simply connected sets for classification purposes. We want to show that simply connected sets can be used for any classification task. However, not all methods of classification using the invex function produce simply connected sets. We discuss in detail the condition when the formed set is simply connected in Appendix E.

#### 3.2.1 Binary Connected Sets

It can be formed by our GC-GP method in Section 3.1.1 and the Invertible method in Section 3.1.2. Invex function with finite minima creates two connected sets, one simply connected set (inner) and another connected set (outer). If the function is monotonous, it creates two simply connected sets (that are not bounded). We can consider the monotonous invex functions as having centroid/minima at infinity. Furthermore, we can create a binary classifier with connected sets as:

$$y_{cluster} = \sigma(-f_{II-NN}(\mathbf{x}))$$

Here, $\sigma(x)$ is a threshold function or sigmoid function. $y = 0$ represents the outer set and $y = 1$ represents the inner set.

Such binary connected sets can be used to determine the region of a One-vs-All classifier and we can form complex regions of classification using multiple such binary connected sets. In this paper, however, we create only $N$ sets for $N$ classes and use $ArgMax$ over class probabilities, which does not create all connected sets classifier despite each classifier having connected decision boundaries.

#### 3.2.2 Multiple Connected Sets

Multiple simply connected sets are created by the nearest centroid classifier, linear classifier or by linear decision trees. However, the connected sets are generally convex and can be mapped to arbitrary shapes by an inverse function. Hence, if we use this in reverse mode, we can transform any space into latent space and then into simply connected sets where the decision boundaries are highly non-linear in the input space. Here, the distance or linear function is convex. Hence, the overall function is invex for each region or node. This is similar to the homeomorphism of connected space as mentioned in Appendix I. Figure 5 shows the

multi-invex classifier using simply connected set leaf nodes. In this paper, we refer to *Multiple Connected Sets* based Classifier by *Multi-Invex Classifier*.

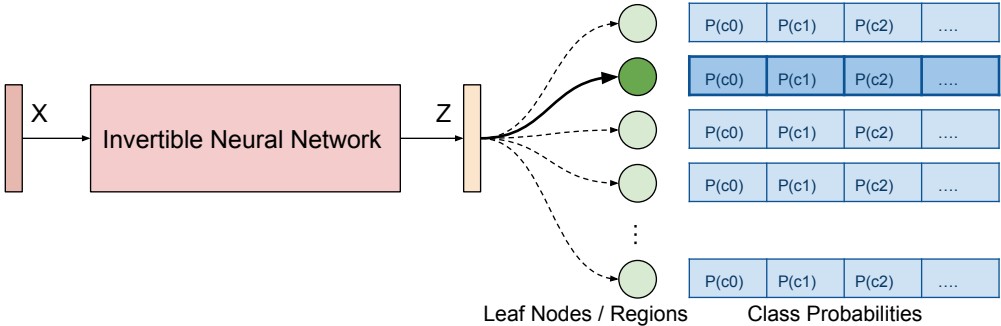

Figure 5: Network Diagram of Multi-Invex Classifier. The Leaf nodes are either produced by a linear decision tree or by the nearest centroid classifier.

We find that ArgMax over multiple convex functions or multiple invex functions does not produce multiple connected sets. Similarly, the Gaussian Mixture Model with gaussian scaling and different variance also does not produce multiple connected sets. This is explained in detail in Appendix E.

---

**Algorithm 2:** Multi Invex Classification - PyTorch like pseudocode

---

```
# X: is the batch of dataset with N Dimensions: shape[Batch Size, D].
# weight: has shape[D, Num Regions], bias shas shape[Num Regions]
# class_prob: has shape[Num Regions, Num Classes], inverse_temp is an scaler
# f_iNN: is an invertible neural network model with trainable parameter.
# f_connected: is a ConnectedClassifier with 'args' representing suitable arguments
# Composing creates multiple invex function with multiple connected decision boundary

class ConnectedClassifier():
    def forward(self, x, weight, bias, class_probs, weight_type, inverse_temp, hard=False):
        if weight_type == 'linear':
            x = torch.matmul(x, weight) + bias
        elif weight_type == 'euclidean':
            x = - (torch.cdist_normalized(x, weight) + bias)
        x = x*torch.exp(inverse_temp)
        ## Here, x is the scaled metric that creates voronoi diagram after argmax.
        if hard:
            x = torch.softmax(x*1e9, dim=1) ## equivalent to argmax + to_one_hot
        else:
            x = torch.softmax(x, dim=1)
        return x.matmul(class_probs)

z = f_iNN(X)
y = f_connected(z, *args)
```

---

**Why do we need Multi-Invex Classifiers ?**  We know that Voronoi diagrams with l2-norm from centroids (or sites) produce convex sets. Such sets can be used to partition the data space into multiple regions and the regions can be assigned class probability for classification tasks (we experiment with this in Section 4.1). Although it is possible to partition the data space into convex regions, it is inefficient due to the limited structure of convex partitioning. Furthermore, Voronoi diagrams produced by l1-norm or by adding a bias term in the output l2-norm can create non-convex partitioning, but still, the partitioning is limited.

Moreover, in general, Voronoi diagrams can even produce disconnected regions (Klein & Wood, 1988) which is not useful for our application. There also have been works on using convex metric functions for Voronoi diagrams (Chew & Dyrsdale III, 1985; Ma, 2000), however, the works are limited to 2D and 3D cases and for a limited number of convex metrics which is insufficient for general application to N-D data points and with learnable convex function. Hence, using an invertible function along with dot-product or l2-norm Voronoi partitioning is suitable for creating multiple connected set classifiers.

Furthermore, Voronoi diagrams have a geometric stability property (Reem, 2011): a small change in the centroids (or sites) creates a small change in the shape of the Voronoi cells. Similar stability holds when some centroids are added or removed; the change in the Voronoi partitions is also small. This plays a crucial role in local function morphism using connected classifiers.

**Interpretability:** If we are to use $z = f_{backbone}(x)$ and $y = f_{classifier}(z)$ then classification $y$ can have different variations of interpretability depending on models used for backbone and classifier. Our method starts with Voronoi partitioning based classification which we call Connected Classifier. It partitions given input space into multiple connected sets and assigns class probability over the regions. Such partitions are however used with softmax for training, but for testing, we can do hard region assignments and hard class output. The partitions and the centroid of the partitions are generally easy to interpret.

However, such classifiers lack the highly non-linear decision boundary that we seek. Hence, our choice is to use an invertible backbone to create a non-linear connected set based classifier. As an invertible function does not fold the space but only morph, it can map every point in input space to latent space and vice versa. This allows us to consider the invertible backbone + connected classifier as a whole in terms of input space and provides high interpretability. The centers on the latent space as well as the decision boundaries can be mapped to the input space.

This is not the case for ordinary NN backbone; the model is ambiguous in terms of input-output mapping and can not be easily used in reversed direction. Hence, using an ordinary NN backbone strips away the interpretability aspect. Furthermore, if we are to use an invertible NN backbone along with the MLP classifier, we do not gain any advantage of interpretability as the MLP classifier itself is hard to interpret and the neurons are highly dependent on each other. Although we can use a linear classifier with an invertible backbone, such a classifier allows only N regions for N classes which might not be optimal for datasets with disconnected classes.

Our approach of using centroids in data space and representing a region using a center matches the concept of Prototype (Rdusseeun & Kaufman, 1987; Biehl et al., 2016) generally used in Explainable AI (Chen et al., 2019; Nauta et al., 2021). However, we do not choose exact data points as centroids but are learned, which makes it different from the prototype, but serves the same purpose of explaining a region of data points using a single concept/region/centroid. We can find the medoid data point of a region which can serve as a Prototype and allow us to interpret the contents in a region, however, if we change the centroid to the medoid, the decision boundary also changes accordingly. It is also possible to use the concept of Prototype and Criticism (Kim et al., 2016) for Network Morphism and to improve connected set based classification, however, we do not experiment with it.

## 4 Experiments

Our experiments mostly consist of a comparison of two of our methods separately with convex and ordinary neural networks. We are unable to make a direct comparison between our two methods due to their fundamental differences in architecture and type of classification performed. In the experiments, we first compare our GCGP method with Convex and Ordinary Neural Networks on multiple datasets. Secondly, we compare our Multi-Invex Classification method with Ordinary Neural Network using different settings on multiple datasets.

### 4.1 Experiments on toy datasets

**Invex function for classification**  To compare the capacity of the invex function to classify connected sets, we experiment on a 2D spiral dataset which is known to have two connected sets as classes. In Table 2 we compare the classification accuracy between Linear, Convex, Ordinary and Invex Neural Networks. We find that our Basic Invex Network can not completely classify the toy classification dataset but composing it one more time helps to achieve 100% accuracy. Furthermore, we show that it can be easily classified by an invex neural network using the invertible method with a similar number of neurons.

Furthermore, we compare visually the convex set classifier and connected-set classifier using toy datasets. The visualization of the decision boundary learned is shown in Appendix C.4. We also consider using the invex function for defining a local region of certainty for robust prediction and rejection of outside samples in Appendix H.

**Multi-convex set vs multi-invex set classification** We compare the use of a multi-invex classifier as compared to multi-convex for a difficult toy 2D classification task. If we use an invertible backbone for the multi-convex classifier, we get a multi-invex classifier. The task is designed to represent disconnected classes for the same class and consists of non-linear class decision boundaries. Figure 6 shows that a convex set based classifier can classify the dataset correctly given a large number of regions, however, connected set based classifier can classify with less number of clusters with more non-linear regions.

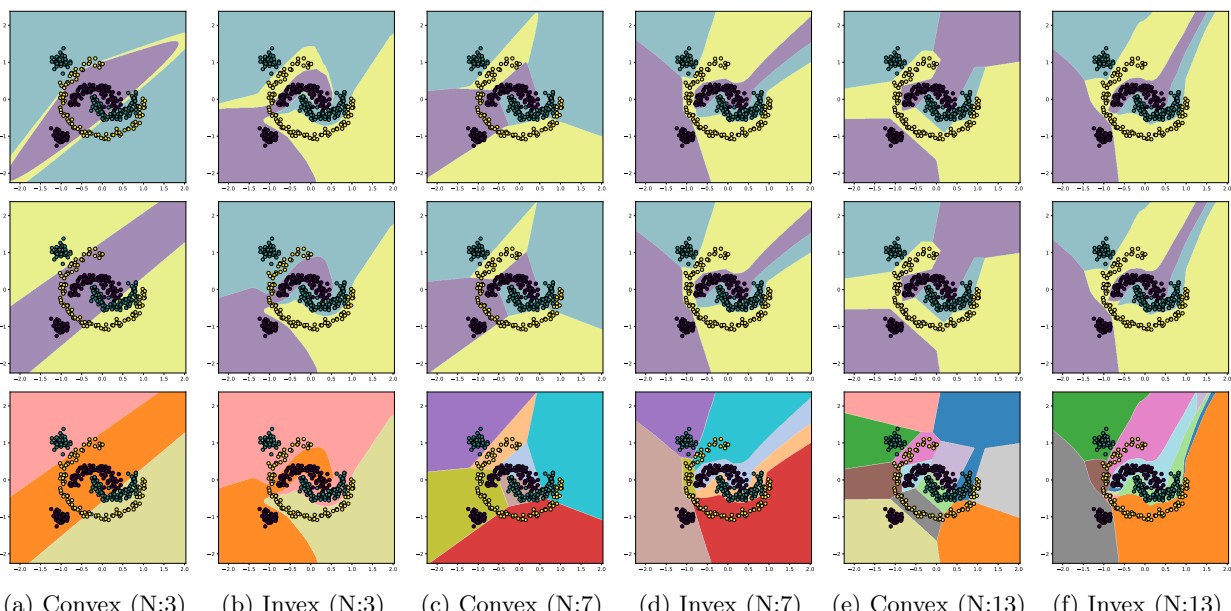

| (a) Convex (N:3) | (b) Invex (N:3) | (c) Convex (N:7) | (d) Invex (N:7) | (e) Convex (N:13) | (f) Invex (N:13) |

Figure 6: Decision boundary of multi-connected classifiers: multi-convex and multi-invex. **(Top Row)** shows the decision boundary learned by softmax over regions, which is difficult to interpret. **(Middle Row)** shows the decision boundary using hard classification i.e. each region is labelled as one of the classes, which makes the classification more interpretable. **(Bottom Row)** shows the regions of classification represented by each neuron of l2-norm based connected classifier. We can simply remove regions containing no data points. *Zoom in the diagram for details.*

**Network Morphism: Adding and Removing Regions**  We extend the experiments on a toy classification problem for conveying the ease of understanding the connected set/region based classifiers. We can easily add new regions without affecting previous regions much and simply assign a class output to them. Furthermore, we can also remove regions that do not add benefit to the classification task. The regions are linked with the centroids, which are individual neurons in the connected classifier layer. We demonstrate the application of network morphism in Figure 7 using a 2D toy classification dataset. The dataset consists of 5 non-linear clusters and 3 classes. Although we can visualize the process in 2D, the underlying mechanism

remains the same for higher dimensions as well. In higher dimensions, we can create similar locally activating neurons (or cluster centroids representing some locality) which can be used to morph the classifier without causing global function changes.

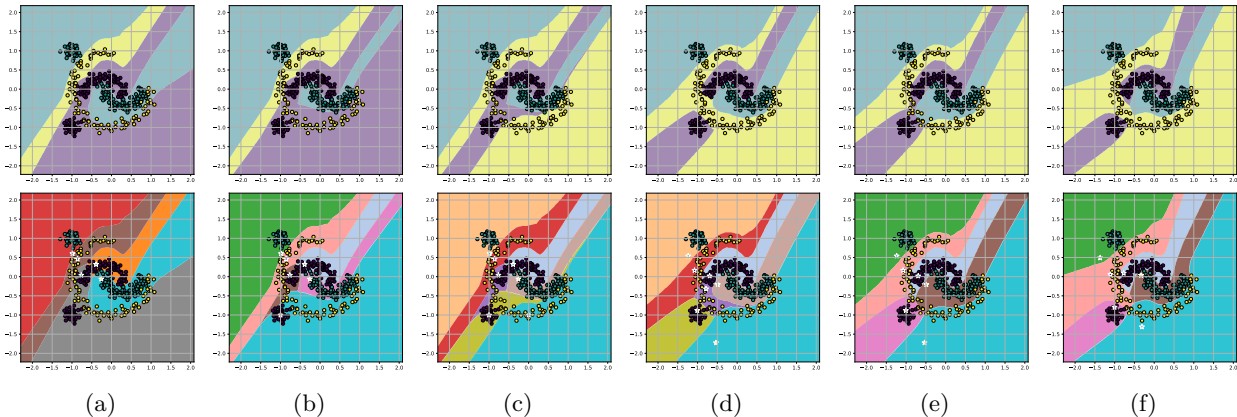

|  (a)  |  (b)  |  (c)  |  (d)  |  (e)  |  (f)  |

Figure 7: **(Top Row)** shows the classification decision boundary learned by the multi-invex classifier with hard (using argmax) over regions. **(Bottom Row)** shows the decision boundary of regions using argmax. The indices (a) - (f) represent different stages of manual network morphism. **(a)** An multi-invex classifier trained with 5 regions. It fails to partition the dataset correctly and performs poorly. **(b)** Adding a new center at (-1, -1) with violet class. **(c)** Adding a new center at (0, -1) with yellow class. **(d)** Fine-tuning the classifier which gets high classification accuracy. **(e)** Removing poorly partitioned center at (-0.78, -0.32) with yellow class. **(f)** Fine-tuning the classifier after removal. *Zoom in the diagram for details.*

Table 2: Accuracy on various Datasets with different Architectures using binary classification. The accuracy on MNIST and FMNIST are summary of 1-vs-all classifier from Table 6, 7, 8. For visualization on the synthetic Classification 1 dataset, check **Appendix F**. The **bold** numbers represent the best results and the blue numbers represent the second-best results. The color gray represents the Invex function using the invertible method. This row is not directly comparable to others due to different neural architectures. The *Basic Invex* row represents the invex function constructed using Proposition 2, the *Invertible (composed)* row represents a modification of Basic Invex model for 1 iteration using Proposition 1. *Invex (invertible)* row represents the Invex function constructed with Invertible Neural Network and Convex Cone.

| Architecture | Dataset (MLP) | | Dataset (CNN) | |
|---|---|---|---|---|
|  | Classification 1 | MNIST | MNIST | F-MNIST |
| Linear/Logistic | 72.0 | 90.79 | - | - |
| Convex | 82.5 | 96.83 | 94.68 | 81.27 |
| Ordinary | **100.0** | 97.09 | **98.08** | 87.76 |
| Basic Invex **(Ours)** | 96.25 | **97.61** | 97.85 | **87.8** |
| Invex (composed) **(Ours)** | **100.0** | - | - | - |
| Invex (Invertible) **(Ours)** | 100.0 | 97.49 | 98.82 | 89.80 |

## 4.2 Experiments on Large Datasets

**One-vs-All classification on Binary Connected Classifiers.** We also experiment on larger-scale datasets, MNIST, and F-MNIST. In Table 2 we compare our GC-GP method with other methods using argmax over multiple (connected) binary classifiers on MNIST and FMNIST datasets. This is due to the limitation of our GC-GP method which can only output a single variable. Although such classification is not generally used in Neural Networks, we find it abundantly in other ML algorithms. This is an inefficient way to create classifiers, however direct comparison is not possible otherwise. In this experiment, we use similar (almost similar) architecture for comparison between Convex, Ordinary and Invex Neural Networks. We also test the invex function using an invertible backbone for relative comparison, however, direct comparison

is not possible due to architectural differences. The hyperparameter $\lambda$ represents regularization constant for Projected Gradient Penalty of GC-GP (See Algorithm 1). It is chosen low for stable training and high for better gradient constraining. We use $\lambda = 2$ on *Basic Invex* and *Invex (composed)* experiments for all datasets. We settle at this specific value with trial and error. Better tuning of $\lambda$ can result in better accuracy. We find that invex function-based classifiers perform similarly to ordinary neural networks on the given datasets. This might be due to the simple clustered nature of classes in input space which is confirmed by the 2D manifold visualization of the invex classifier in Appendix G.

We can not be certain if a function is invex or not in a high-dimensional space like MNIST when using the GC-GP method. To verify that our function is invex we test for constraints on all training and test datasets as well as on 1 Million random points. It is found that in F-MNIST our classifiers follow our invexity rule on $> 99\%$ of data points and $> 99\%$ of random points. In the MNIST dataset, for both MLP and CNN architecture, some classifiers have fewer percentages of points that follow our invex rule. The details of the experiments and the percentage of points following our invexity rule are mentioned in the Appendix section C.4.

Table 3: Accuracy on various datasets using various architectures. The blue numbers represent Accuracy using hard classification as per nearest region/classifier for Connected Classifier. *iNN* represents invertible Neural Network as backbone whereas *NN* represents ordinary Neural Network for backbone. *Connected Classifier* represents *our method* of classification and *MLP Classifier* represents ordinary 2 layer MLP for classification task. The settings using *iNN + Connected Classifier* and *iNN + Linear Classifier* produce connected set based classification. (**\***) Represents model used for interpretation in Table 4

| Architecture | Dataset (MLP) | Dataset (CNN) | | | |
|---|---|---|---|---|---|
| | MNIST | MNIST | F-MNIST | C-10 | C-100 |
| iNN + Connected Classifier | 96.68 | 98.87 | 89.58 | 84.45 | 52.58 |
| | 96.77 | 98.78 | 89.49 | 84.23**\*** | 51.85 |
| iNN + MLP Classifier | 96.81 | 98.26 | 89.61 | 84.77 | 56.91 |
| NN + Connected Classifier | 97.55 | 99.4 | 90.38 | 85.93 | 50.74 |
| | 97.54 | 99.34 | 88.1 | 85.79 | 50.39 |
| NN + MLP Classifier | 97.38 | 98.89 | 90.58 | 86.02 | 53.08 |
| iNN + Connected Classifier | 96.83 | 98.38 | 89.35 | 83.95 | 52.44 |
| (Regions = Classes) | 96.85 | 98.43 | 88.43 | 83.81 | 51.54 |
| iNN + Linear Classifier | 96.61 | 98.24 | 88.97 | 84.71 | 56.08 |

**Multi-connected set classifier**  We compare the classification capacity of a multi-connected set classifier with an ordinary neural network. Here, we use invertible architecture along with the same architecture without invertibility for comparison. The details regarding the classification method are mentioned in Methodology Section.

Here, we directly compare the classification capacity of ordinary neural networks with connected-set-based classifiers. The classification accuracy of different models on MNIST, F-MNIST, CIFAR-10 and CIFAR-100 datasets are in Table 3. Furthermore, we also use node-based classification on top of the ordinary backbone for a fair comparison. We use an invertible or ordinary backbone combined with a connected or MLP(disconnected) classifier for a detailed comparison. To test the benefit of a multi-connected-set classifier we also test using an invertible backbone and linear classifier. The experiments show that our connected classifier performs poorly as compared to MLP or Linear classifier, however, using a linear equivalent connected classifier, i.e. using the number of regions equal to the number of classes, we find that using multiple regions helps in accuracy. This gap in performance can be credited to the poor optimization of the connected classifier. We initialize the ordinary neural network using spectral normalization (SN) which improves the performance of the ordinary neural network on invertible architecture. For all experiments, we use an l2-norm based connected classifier except for the CIFAR-100 experiment where we use a dot-product based connected classifier. Furthermore, we classify using the multi-invex classifier on a 2-D manifold in input space (see Appendix G). Such classifiers have lower classification accuracy but output the manifold in 2-D which can be plotted similarly to UMAP (McInnes et al., 2018).

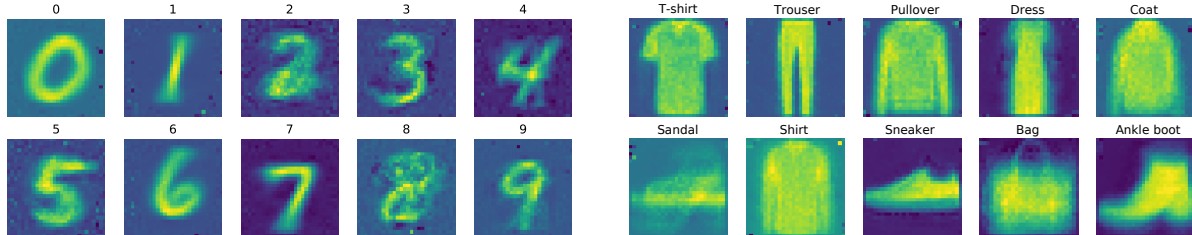

Figure 8: **Left**: Centroids of 10 classifiers (in input space) for MNIST. **Right**: Centroids of 10 classifiers (in input space) for Fashion-MNIST. These centroids visualization is done with models using Invertible Residual MLP trained with 1-vs-all classification and not from Table 2 or 3.

**Interpretation of Regions/Neurons**   The centroid of each classifier of the One-vs-All classification can be visualized since the centroid represents the maximal class point. The visualization helps us to interpret where the neuron is focusing in a local region (or compact 1-connected set) around the centroid. We observe the centroids of 10 MNIST classifiers and 10 F-MNIST one-vs-all classifiers per class to have centroids as shown in Figure 8. We transform the center of the cone created by binary connected classifier layers by using an invertible neural network in the reverse direction. This gives us the centers in terms of input space which can be visualized as an image.

Furthermore, the centers of the Multi-Connected Set based classifier can also be visualized as shown in Table 4. We find that centers do not exactly represent the data points, and are not explainable in terms of input space. The centers on the table show that the region contains noisy points similar to adversarial examples (Goodfellow et al., 2014) and can produce output with high class probability. We can similarly perform an analysis of each neuron representing a region in terms of accuracy and visualize the medoid of the data points as well as an example nearest to the centroid in each region.

## 5   Limitations

Invex function using gradient constraint depends on GC-GP, a robust method, however, it can not guarantee if the function is invex or not. When we experiment on 1-Lipschitz Constraint in Table 5, we find that our method successfully constrains properly for different values of lambda($\lambda$). However, the constraint is not followed properly for some $\lambda$ and some experiments. This observation also follows the projected gradient constraint used for the invex function. Table 6, 7, 8 show the percentage of points following the constraints, where we find that there are points that do not satisfy the constraints. This might be a problem for certain cases where invexity needs to be guaranteed. Even if the constraint is satisfied for all data points by the learned function, we need to confirm if it satisfies for all possible points. In such a case, we can theoretically argue that some invex function with the same gradients on those data points satisfies the constraint for all possible points. Furthermore, we need robust mathematical proofs for our gradient constraint method of constructing the invex function. Our intuition is easily understood in 3D visualization. However, we can only interpolate the idea of equations using general operations like the dot product, norm and inequality for any dimension. The connected classifier used in Multi-Invex classifier experiments shows poor performance as compared to the MLP classifier which is mainly due to poor optimization of the model when used with a connected classifier. This gap in performance can be analysed and improved. We are also unsure if the partitions of binary connected classifier and multi-invex classifier are overfitting for high dimensional datasets. We consider regularizing invertible functions for simpler decision boundaries and applying the concept of Prototype and Criticism used in Explainable AI to improve the Connected Classifiers for our future work. See Appendix J for future works.

Table 4: Interpretation of connected regions of Multi-Invex (multiple connected set based) classifier on CIFAR-10. The **Center** represents the center used by the region in Voronoi Diagram to create a decision boundary, it is not interpretable, however, belongs to the same region. Z-space represents observations in latent space after invertible transformation and X-space represents observations in input space. **Medoid** examples represent the medoid of data points belonging to the region. **Nearest** examples show the nearest example to the *Center* for the region. **Class**, name and index, represents the class the region/neuron belongs to and RegionId represents the index of the region/neuron among 100 used in the experiments; the remaining region/neuron do not contain any test points and can be removed. The region contains **Num Points** total samples with **Correct** number of samples represented by **Accuracy** in percentage.

| Class RegionId | 2 (2) bird | 7 (17) horse | 1 (21) automobile | 4 (24) deer | 5 (25) dog | 6 (26) frog | 3 (33) cat | 9 (49) truck | 0 (50) airplane |
|---|---|---|---|---|---|---|---|---|---|
| Num Points | 424 | 986 | 974 | 1043 | 13 | 1023 | 67 | 5 | 953 |
| Correct | 366 | 874 | 911 | 851 | 7 | 898 | 26 | 2 | 809 |
| Accuracy(%) | 86.56 | 88.74 | 93.53 | 81.59 | 53.85 | 87.88 | 40.30 | 40.00 | 84.89 |

| Class RegionId | 9 (59) truck | 1 (61) automobile | 2 (62) bird | 5 (65) dog | 7 (77) horse | 2 (82) bird | 8 (88) ship | 0 (90) airplane | 3 (93) cat |
|---|---|---|---|---|---|---|---|---|---|
| Num Points | 1024 | 3 | 412 | 972 | 9 | 59 | 1002 | 94 | 937 |
| Correct | 907 | 3 | 341 | 725 | 2 | 48 | 916 | 77 | 653 |
| Accuracy(%) | 88.57 | 100.0 | 82.77 | 74.59 | 22.22 | 83.05 | 91.52 | 81.91 | 69.80 |

# 6 Conclusion

In this paper, we introduced a new type of Neural Network called Input Invex Neural Network (II-NN). We present two methods to create an invex function with neural networks. Experiments show that II-NN has comparable performance to that of Ordinary Neural Networks in the classification task. We relate the concept of a simply-connected set with the invex function and show that multiple simply connected decision boundaries can perform any classification task.

Furthermore, we relate the concept of simply connected decision boundaries with interpretability. Knowing the centroid of the set, the data points in it and the maximum value it represents can be useful for multiple applications such as activation visualization and neuron interpretation. We also exploit the geometric stability property of the Connected Set Classifier to perform Network Morphism on toy datasets.

Although we use GC-GP for constructing the invex function, it can be used for other tasks such as on WGAN (Arjovsky et al., 2017) or modified for other constraining problems. Connected regions can also be used for other applications than we experiment with; II-NN will be useful in such cases.

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

# Appendices

## A  Gadient-Clipped Gradient Penalty

K-Lipschitz constraint of Neural Networks has been important for multiple works such as WGAN and iResNet. WGAN-GP (Gulrajani et al., 2017) regularizes the Lipschitz constant of the data points to be some value (eg. K=1). This method has two drawbacks. Firstly, it cannot exactly constrain the gradient to be precisely K-Lipschitz as it is added to the loss term and constrained via gradient descent. This is shown in the experiment section in Table 5. It is because, in many training examples, the gradient from the criterion is opposite to the gradient from the gradient-penalty, which does not allow the desired Lipschitz constant. Secondly, the constraint adds the loss if the local Lipschitz constant at some points is less than desired. According to the definition of the Lipschitz constant, the local Lipschitz value can be any below the maximum K specified. Similarly, WGAN-LP (Petzka et al., 2018) regularizes only if the local Lipschitz constant at a point is greater than the specified K. It solves the second problem with WGAN-GP, yet it faces the same first drawback. Spectral Normalization (Miyato et al., 2018) is another robust method for constraining the K-Lipschitz constant of the Neural Network. It constrains the Neural Network at the functional level, i.e. constraints the weights. The problem with this method is that it constrains the upper bound of the Lipschitz constant globally.

Although WGAN-GP and WGAN-LP constrain the function globally, they can be modified to constrain locally as well. Since these methods constrain the gradient of the input w.r.t output, it can constrain each input point to a different magnitude of the gradient, i.e. to a specific local Lipschitz constant. The Spectral Normalization (Miyato et al., 2018), however, cannot constrain the local K-Lipschitz or input gradient. To construct an invex function, we have a requirement to have local K-Lipschitz constraint or input gradient constraint guaranteed as shown in Proposition 1.

### A.1  How our method solves the problem

Since we develop a method for constructing invex function depending on the projected input gradient constrained neural network, we also engineered a method to impose such constrain on Neural Networks. Our method (GC-GP) improves on the drawbacks of previous gradient constraint methods. The details are discussed in Section 3.

To summarize, we use two components to constrain local gradient constraints. First, we apply Gradient Penalty (GP) according to the Gradient Constraint required (Gradient Magnitude in the case of K-Lipschitz and Projected Gradient Constraint in case of constructing invex functions). It is known that the criterion can also oppose the Gradient Constraint, which will not constrain the gradient properly. To solve this, we apply gradient clipping by intercepting the gradient at the output neuron. We zero out the criterion gradient at those points where gradient constraint is being violated. This allows us to achieve two goals, fitting neural network to the data and constraining gradient (make function K-Lipschitz or invex). The pipeline for gradient constraint is shown by Figure 4. The pseudocode for GC-GP used in the invex function is shown in Algorithm 1.

### A.2  Limitations of our method

Although our method (GC-GP) improves upon GP and LP, it can not guarantee the constraint is satisfied, it only regularizes towards that constraint. Although we experimentally show (in Section B) that the majority of data points satisfy our constraint, it is still not a theoretical guarantee. This might limit the use of GC-GP for some theoretical works and proofs.

# B    Lipschitz Constraint : Experiment

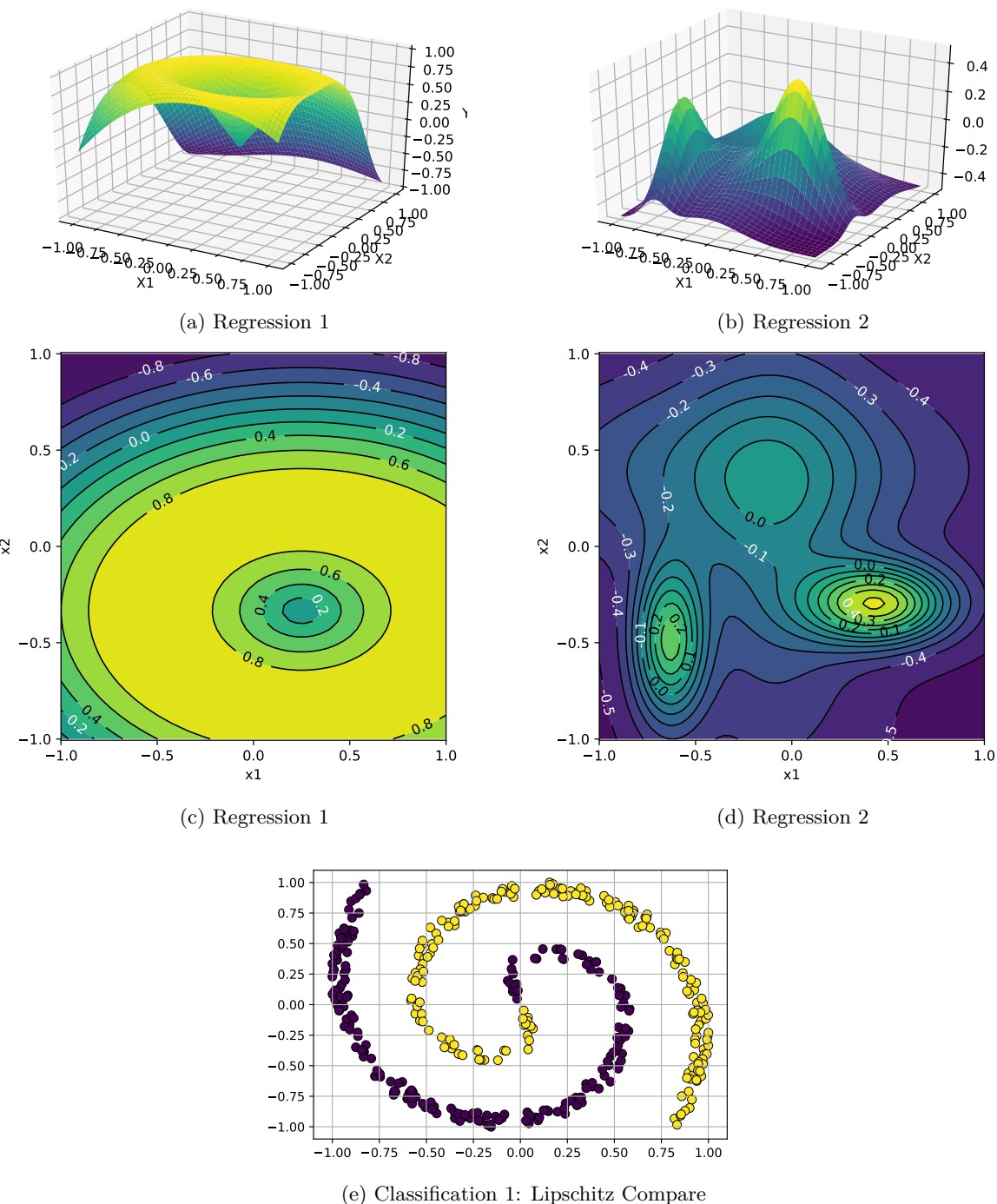

(a) Regression 1

(b) Regression 2

(c) Regression 1

(d) Regression 2

(e) Classification 1: Lipschitz Compare

Figure 9: (a) and (c) are the 3D plot and Contour plot of Regression 1 dataset respectively. Similarly, (b) and (d) are the 3D plot and Contour plot of the Regression 2 dataset respectively. (e) is the Classification 1 Dataset (Spiral). These datasets are used for the Lipschitz constraint experiment as well as invex function experiments. The source code for the datasets is available in the supplementary material.

We conduct detailed experiments for comparing the Lipschitz constraint of various methods in Table 5. It can be observed that our method (GC-GP) achieves relatively high performance while maintaining the Lipschitz

constant close to the target ($K = 1$). In the Classification 1 dataset, our method has a Lipschitz constant close to one (1) and does better with $\lambda = 3$. Although GC-GP can not guarantee the target Lipschitz constant, it can be seen from the experiment that it achieves near-perfect constraint.

Firstly, we experiment with three toy datasets for comparing the Lipschitz constraint on regression and classification with various methods. These are: two 2D regression and a 2D classification dataset. The Regression 1 and 2 datasets consist of 2500 and 5625 points on a grid respectively whereas the Classification 1 dataset is a 2D spiral data consisting of 400 data points. For regression, we test on Mean Squared Error (MSE) and for classification, we test on Binary Cross Entropy (BCE) and Accuracy on the training data. We compare constraint methods: Gradient-Penalty (GP) (Gulrajani et al., 2017), Lipschitz-Penalty (LP) (Petzka et al., 2018), Spectral Normalization (SN) (Miyato et al., 2018) and Our Method (Gradient Clipped Gradient Penalty, GC-GP) for 1-Lipschitz function. The metrics and Lipschitz constant of function learned are shown in Table 5. The dataset used for Regression 1, 2 and Classification 1 are in Figure 9.

We use the scaled sigmoid function for the final layer of the Spectral Normalized Neural Network during classification. The sigmoid is scaled by 4 such that its Lipschitz constant is 1. Thus, the Lipschitz constant of the overall function is unchanged by the sigmoid activation. The experiments on Table 5 is conducted on Neural Network with configuration: (2,10,10,1), where 2 and 1 are input and output dimension respectively. We use ELU (Clevert et al., 2015) activation for regression and LeakyReLU (Maas et al., 2013) activation for classification in intermediate layers. We use sigmoid in the final layer for classification. We train each model for 7500 epochs using a full batch using soft-l1-loss for penalizing the gradient norm (or K-Lipschitz).

| Dataset | Method | Seed | Loss / (Accuracy) | Lipschitz Constant (K) | Minimum Gradient Norm | Time (ms) |
|---|---|---|---|---|---|---|
| Regression 1 | GP ($\lambda = 1$) | A | 0.06793 | 1.37824 | 0.12656 | 4.12±0.58 |
| | | B | 0.06391 | 1.45713 | 0.28021 | |
| | | C | 0.07396 | 1.23643 | 0.30802 | |
| | GP ($\lambda = 3$) | A | 0.11041 | 1.22805 | 0.59818 | 4.12±0.60 |
| | | B | 0.11135 | 1.21324 | 0.55712 | |
| | | C | 0.10442 | 1.18923 | 0.62169 | |
| | LP ($\lambda = 1$) | A | 0.05598 | 1.21801 | 0.01146 | 4.19±0.56 |
| | | B | 0.05524 | 1.25629 | 0.02150 | |
| | | C | 0.05597 | 1.20679 | 0.01799 | |
| | LP ($\lambda = 3$) | A | 0.06335 | 1.10782 | 0.02662 | 4.17±0.56 |
| | | B | 0.06344 | 1.12563 | 0.00471 | |
| | | C | 0.06310 | 1.18670 | 0.01166 | |
| | SN | A | 0.09156 | 1.00014 | 0.12324 | 2.88±0.47 |
| | | B | 0.09139 | 0.99901 | 0.12381 | |
| | | C | 0.09121 | 0.99785 | 0.12192 | |
| | GC-GP ($\lambda = 1$) **(Ours)** | A | 0.08365 | 0.92083 | 0.02074 | 5.60±0.96 |
| | | B | 0.08426 | 0.99231 | 0.01201 | |
| | | C | 0.08185 | 0.92162 | 0.02632 | |
| | GC-GP ($\lambda = 3$) **(Ours)** | A | 0.08874 | 0.88536 | 0.03419 | 6.05±0.77 |
| | | B | 0.08708 | 0.87693 | 0.02011 | |
| | | C | 0.08819 | 0.88356 | 0.01996 | |
| Regression 2 | GP ($\lambda = 1$) | A | 0.02060 | 1.19123 | 0.32234 | 6.06±0.81 |
| | | B | 0.01715 | 1.35935 | 0.25872 | |
| | | C | 0.01786 | 1.31234 | 0.24638 | |
| | GP ($\lambda = 3$) | A | 0.02641 | 1.14505 | 0.540290 | 7.80±0.75 |
| | | B | 0.02131 | 1.13570 | 0.47666 | |
| | | C | 0.03304 | 1.20034 | 0.57999 | |
| | LP ($\lambda = 1$) | A | 0.00530 | 1.12591 | 0.00758 | 6.13±0.78 |
| | | B | 0.00524 | 1.13546 | 0.01011 | |
| | | C | 0.00529 | 1.17633 | 0.00312 | |
| | LP ($\lambda = 3$) | A | 0.00571 | 1.14781 | 0.00490 | 6.15±0.84 |
| | | B | 0.00550 | 1.07732 | 0.02155 | |
| | | C | 0.00546 | 1.07079 | 0.01466 | |
| | SN | A | 0.01774 | 0.45701 | 0.006839 | 3.27±0.54 |
| | | B | 0.01923 | 0.47381 | 0.00768 | |
| | | C | 0.01871 | 0.48544 | 0.00780 | |
| | GC-GP ($\lambda = 1$) **(Ours)** | A | 0.00759 | 0.89684 | 0.00537 | 7.75±0.75 |
| | | B | 0.00720 | 0.89899 | 0.01888 | |
| | | C | 0.00729 | 0.89305 | 0.01133 | |
| | GC-GP ($\lambda = 3$) **(Ours)** | A | 0.00757 | 0.86108 | 0.00964 | 6.05±0.77 |
| | | B | 0.00757 | 0.85756 | 0.01369 | |
| | | C | 0.00826 | 0.85425 | 0.00193 | |
| Classification 1 | GP ($\lambda = 1$) | A | 0.12006 *(100.0)* | 1.67273 | 0.28320 | 2.26±0.45 |
| | | B | 0.11239 *(100.0)* | 1.74431 | 0.37148 | |
| | | C | 0.11637 *(100.0)* | 1.71859 | 0.26768 | |
| | GP ($\lambda = 3$) | A | 0.20353 *(98.0)* | 1.66700 | 0.35362 | 2.60±0.46 |
| | | B | 0.23160 *(97.25)* | 1.49510 | 0.51360 | |
| | | C | 0.19764 *(99.25)* | 1.51147 | 0.03693 | |
| | LP ($\lambda = 1$) | A | 0.01828 *(100.0)* | 1.37528 | 0.0 | 2.66±0.47 |
| | | B | 0.06048 *(97.25)* | 2.25560 | 0.0 | |
| | | C | 0.01934 *(100.0)* | 1.60750 | 0.0 | |
| | LP ($\lambda = 3$) | A | 0.17167 *(96.5)* | 1.58051 | 0.00492 | 2.66±0.46 |
| | | B | 0.054330 *(100.0)* | 1.80411 | $1.72 \times 10^{-7}$ | |
| | | C | 0.050757 *(97.25)* | 2.06773 | 0.0 | |
| | SN | A | 0.43505 *(76.0)* | 0.79094 | 0.17857 | 2.51±0.45 |
| | | B | 0.42814 *(76.5)* | 0.85739 | 0.22571 | |
| | | C | 0.43819 *(76.25)* | 0.95832 | 0.19253 | |
| | GC-GP ($\lambda = 1$) **(Ours)** | A | 0.30689 *(98.0)* | 1.19644 | 0.18211 | 3.55±0.40 |
| | | B | 0.35706 *(84.75)* | 1.03542 | 0.07420 | |
| | | C | 0.29447 *(98.0)* | 1.04589 | 0.130878 | |
| | GC-GP ($\lambda = 3$) **(Ours)** | A | 0.32096 *(96.0)* | 0.97788 | 0.14575 | 3.55±0.40 |
| | | B | 0.35798 *(85.25)* | 0.96265 | 0.07568 | |
| | | C | 0.33111 *(94.75)* | 1.07161 | 0.14411 | |

Table 5: Comparison between various K-Lipschitz Constraint methods. Seeds A, B and C are 147, 258 and 369 respectively. The time taken is measured for all seeds. Our method takes the most time as it needs to scale the penalty and clip the gradient. Spectral Normalization (SN) takes the least time as it does layerwise iterative normalization and does not depend on the data.

## C Invex Function using GC-GP

### C.1 Motivation/Intution for Connected Sets and Invex function

To simplify our problem, we start with a 1D function that is not convex but has only global minima. To our surprise, we found that such a function can actually be decomposed into two sums, a convex part and a locally Lipschitz constrained part as shown in Figure 10. We first develop a constraint that is required for the function to be quasi-convex in 1D (A strongly quasi-convex function in 1D is equivalent to an invex function with a single minima point).

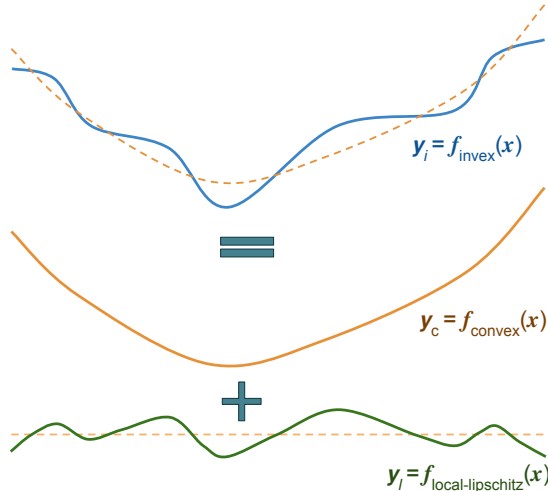

Figure 10: A 1D invex function $f_{invex}(x)$ decomposed into convex $f_{convex}(x)$ and $f_{local-lipschitz}(x)$. This serves as an initial motivation for finding invex function using gradient constraint/local-Lipschitz constraint method.

We found the constraint to be as follows. However, generalizing to a higher dimension was a challenge.

$$\text{sign}\Big(\frac{dy_c}{dx}\Big) \times \Big(\frac{dy_c}{dx} + \frac{dy_l}{dx}\Big) > 0$$

We found that we were looking for an invex function, as it was the only generalization that had a requirement to have only global minima as stationary points. Modifying the above equation for 2D, we find that similar to the local Lipschitz constraint, the local projected gradient constraint should be done by using a dot product instead. In the next section, we present the idea in an organized format, not parallel with the development of the theory and its implementation.

### C.2 Theory of Constructing Invex Function

**Proposition 1** *Let $f : \mathbf{X} \to \mathbb{R}$ and $g : \mathbf{X} \to \mathbb{R}$ be two functions on vector space $\mathbf{X}$. Let $\mathbf{x} \in \mathbf{X}$ be any point, $\mathbf{x}^*$ be the minima of $f$ and $\mathbf{x} \neq \mathbf{x}^*$. If $f$ be an invex function (convex or strongly quasi-convex) and If*

$$\Big(\frac{\nabla g(\mathbf{x}) \cdot \nabla f(\mathbf{x})}{\|\nabla f(\mathbf{x})\|}\Big) + \|\nabla f(\mathbf{x})\| > 0$$

*then $h(\mathbf{x}) = f(\mathbf{x}) + g(\mathbf{x})$ is an invex function.*

**Proof in 1D**: Let us consider a 1D invex function $f(x)$ as shown in Figure 11. If we add the function $f(x)$ with some $g_i(x)$, then we will have a new function $h_i(x)$. The modified function might be invex or not. If it does not change the direction of the gradient at any point with reference to the gradient of the original function $f(x)$, i.e.

$$\nabla h_i(\mathbf{x}) \cdot \nabla f(\mathbf{x}) > 0 \tag{6}$$

then $h_i(x)$ is an invex function. If the modified function changes the direction of the gradient, it would mean that there exists new minima/maxima that are not the minima of the $f(x)$. If we preserve the above inequality, we keep the position of minima intact and modify only those part which preserves the invexity. Those modifications which do not follow the above rule might still be an invex function. For example, if the modification shifts the function in Figure 11 from left to right, it is still an invex function, but the direction of the gradient would be opposing in many input points. In Figure 11, we use $x_1$ and $x_2$ as example points to show the gradient and the inequality. The modified functions: $h_1(x)$, $h_2(x)$ and $h_3(x)$ have the same direction of gradient at all points hence, these are invex function. The modified functions $h_0(x)$ and $h_4(x)$ have different gradient direction at some points. Hence, these functions are not invex. In 1D, the dot product between the original gradient and gradient of the modified function at each point gives a $+ve$ value if the direction is the same. We want all the dot products between gradients to be positive, to preserve invexity while changing the non-linearity of the function. We ignore the inequality at minima ($\mathbf{x}^*$) because the gradient is zero and does not follow the inequality.

This definition is still true in 2D for which the visual proof is shown in the next paragraph. This idea can be extended to n-Dimensional (nD) functions as well. The main goal is to change the existing invex (or convex) function such that the modified gradient still leads to the same global minima. If the projected gradient of the modified function $\nabla h_i(\mathbf{x})$ in the direction of $\nabla f(\mathbf{x})$ is positive at all points, it implies that there is no new maxima/minima created during the modification. This is what preserves the invexity. We extrapolate the idea to nD functions as well but we lack proof for it. However, the idea is intuitive enough to suggest that it most likely holds true for nD as well.

**Extrapolation from convex function:** Let us consider only the nD differentiable convex function. We know that a differentiable convex function has global minima as well as convex contour sets. A convex contour set is a generalization of a 1-connected set. Furthermore, there are two types of convex function: **(1)** with a minimum at finite $\mathbf{x}$ (or local convex function) which has a bounded convex contour set. **(2)** with minima at some infinity ($\infty$) which has an unbounded convex contour set.

We take the property of a differentiable convex function that it has a global minimum and modify the contour sets, with a new function superposition, such that the sets are still connected, i.e. has only the same single global minima, and are more non-linear. Such a general function having only global minima is called an invex function. Our initial axiom is defined by equation (4). If the angle between gradients of the old convex function and modified function is less than $\frac{\pi}{2}$ at all points, then we can say that no new minima have been formed. This is because, if any new minima are formed then around that minima, the gradients of the old function and that of the modified function around the minima would be opposing at some points.

This modification proposition can be applied to an invex function as well as to a simple convex function like a cone. However, our method will prevent the function from having a large change in direction (i.e. angle $\geq \frac{\pi}{2}$), which could still be an invex function. This is tackled by propositions 2 and 3.

*In applications like machine learning, the data points are finite and normalized to a small range. Here, locality might refer to having boundaries in the range of $\mathbf{x}$.*

This statement can be written mathematically.

$$\nabla h(\mathbf{x}) \cdot \frac{\nabla f(\mathbf{x})}{\|\nabla f(\mathbf{x})\|} > 0 \tag{7}$$

Using $h(\mathbf{x}) = f(\mathbf{x}) + g(\mathbf{x})$ and solving this we get,

$$\left( \frac{\nabla g(\mathbf{x}) \cdot \nabla f(\mathbf{x})}{\|\nabla f(\mathbf{x})\|} \right) + \|\nabla f(\mathbf{x})\| > 0 \tag{8}$$

**Proof in 2D:** Consider a simple invex function $f(\mathbf{x})$ as shown in figure 12. It is a simple modification of the sigmoid function. We can make it more non-linear by adding another function $g(\mathbf{x})$. If the modified function $g(\mathbf{x})$ does not change the direction of the projected gradient at any point, then it is an invex function.

$$\nabla h(\mathbf{x}) \cdot \nabla f(\mathbf{x}) > 0 \tag{9}$$

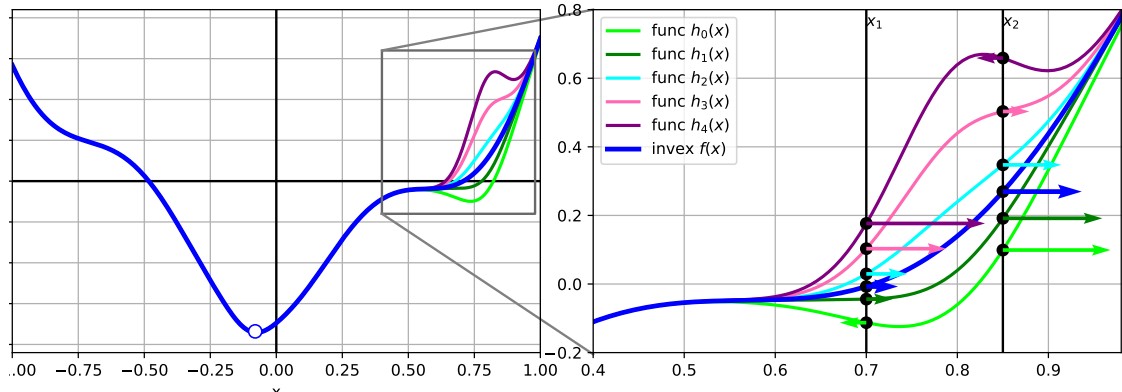

Figure 11: A 1D function $f(x)$ with modification by various $g_i(x)$, forming modified function $h_i(x)$.

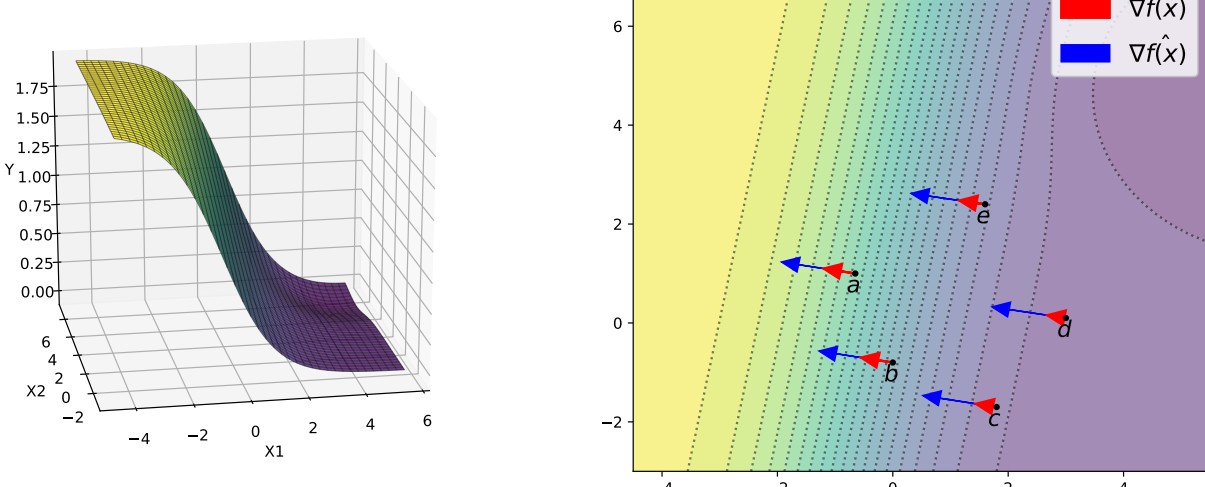

Figure 12: Initial invex function. This function is modified and checked at points a-e to find if the function is invex or not. The check actually happens at all points in the algorithm. However, here we use these 5 points to convey the property of change in direction of the gradient and its relation with invex function.

We present 4 modifications to the function in Figure 12 which shows various cases to prove our proposition. We choose 5 points (a, b, c, d and e) which show condition being satisfied or not being satisfied by various modifications.

**Modification 1:** The Figure 13 shows $h(\mathbf{x}) = f(\mathbf{x}) + g(\mathbf{x})$. The modification has the same global minima, i.e. no new minima/maxima have been created. The modified function satisfies the condition in Equation 9 at all points. It can be seen that the projected gradient all have positive values. It can also be seen on the contour plot that there are no new minima or maxima. Hence, the modification is still an invex function.

**Modification 2:** The modification as shown in Figure 14 is similar to Modification 1. The modified function is still an invex function (same reason as Modification 1).

**Modification 3:** The modification as shown in Figure 15 is not an invex function. It has a new maximum as compared to the original function, as can be seen on the contour plot. This is reflected by the violation of Equation 9. If we look at point a, then we can see that the projected gradient of modified function $h(\mathbf{x})$

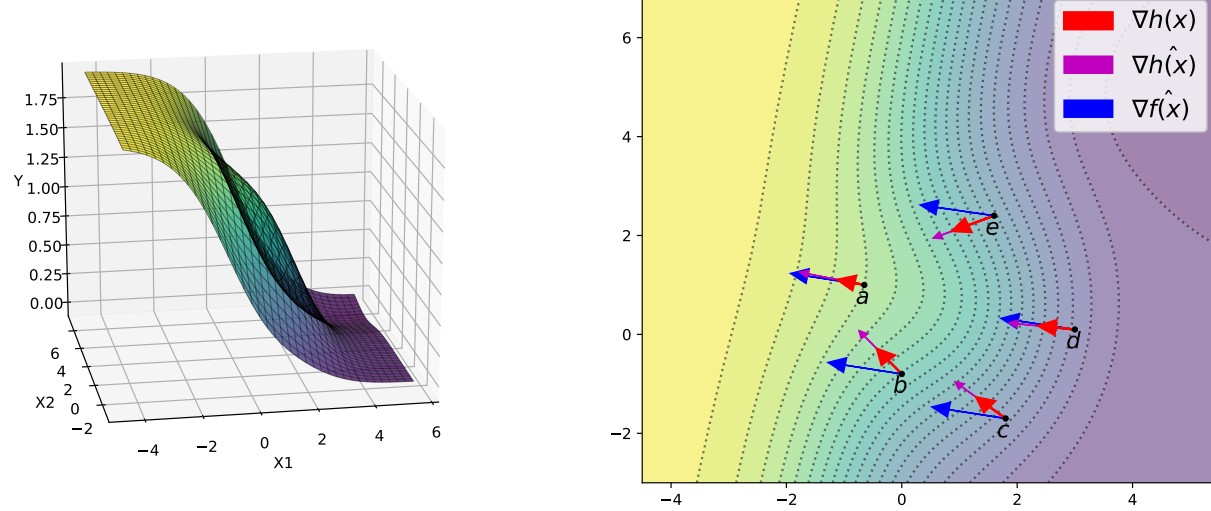

Figure 13: Modified invex function 1, also an invex function.

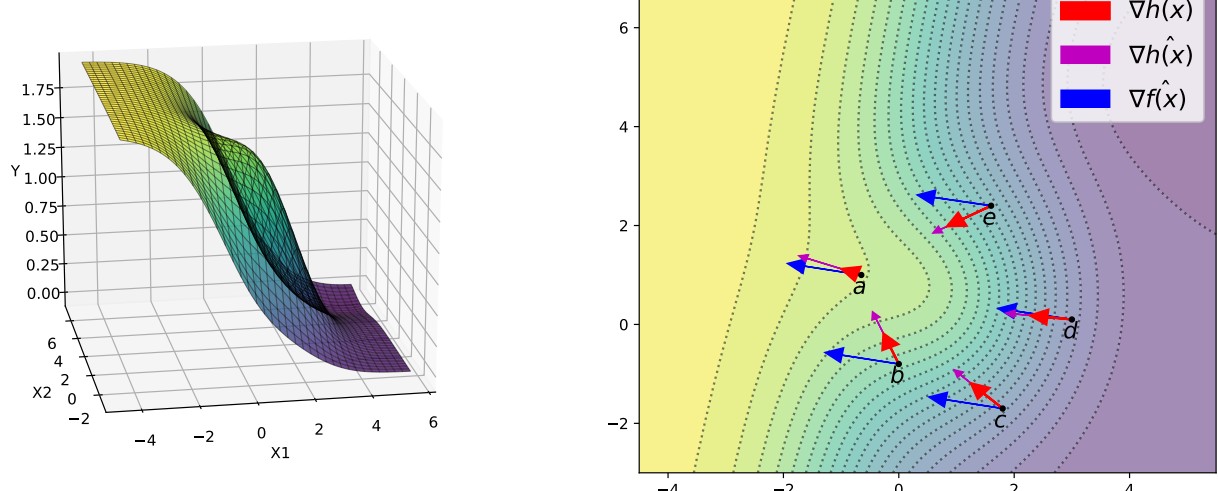

Figure 14: Modified invex function 2, also an invex function.

on $f(\mathbf{x})$ is negative. Since it violates our condition, we are not sure that it is an invex function. Hence, our condition is violated if there is a new minima/maxima, which is predicted by the projected gradient.

**Modification 4:** The modification as shown in Figure 16 is also not an invex function. It has a new minimum as compared to the original function. It can be seen on the contour plot as well. The modified function violates the Equation 9 constraint at point d (and points around it). Hence it may not be an invex function. In this case, it is not an invex function. But we can not say that the function is not invex if it violates Equation 9.

**Proposition 2** *Let $g : \mathbf{X} \to \mathbb{R}$ be a function on vector space $\mathbf{X}$, let $\mathbf{x} \in \mathbf{X}$ be any point, $\mathbf{x}^*$ be the minima of $g$ and $\mathbf{x} \neq \mathbf{x}^*$. If*

$$\nabla g(\mathbf{x}) \cdot \frac{\mathbf{x} - \mathbf{x}^*}{\|\mathbf{x} - \mathbf{x}^*\|} > 0$$

*then $g$ is an invex function.*

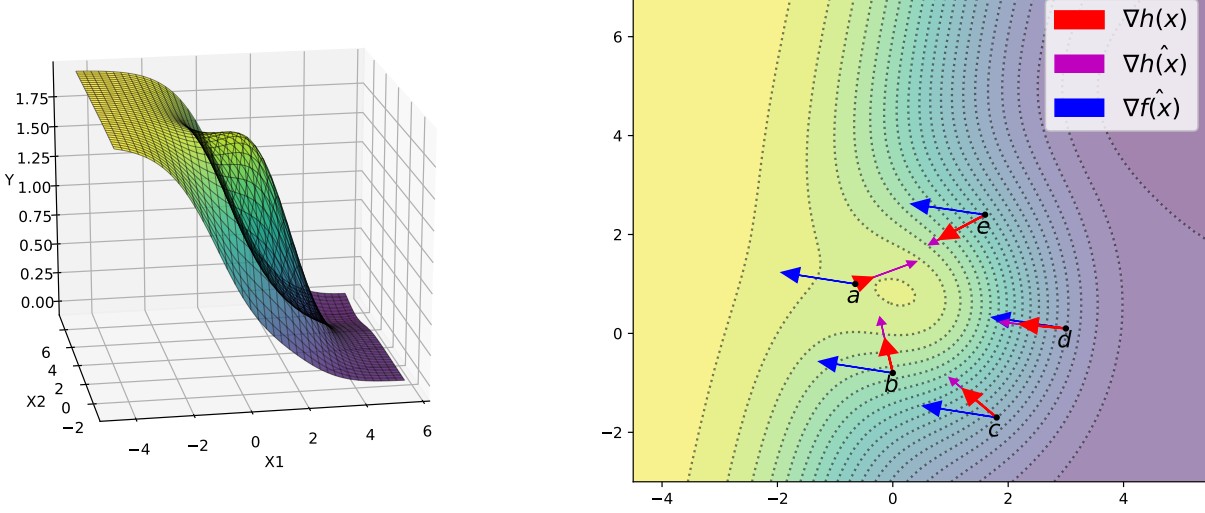

Figure 15: Modified invex function 3, not an invex function.

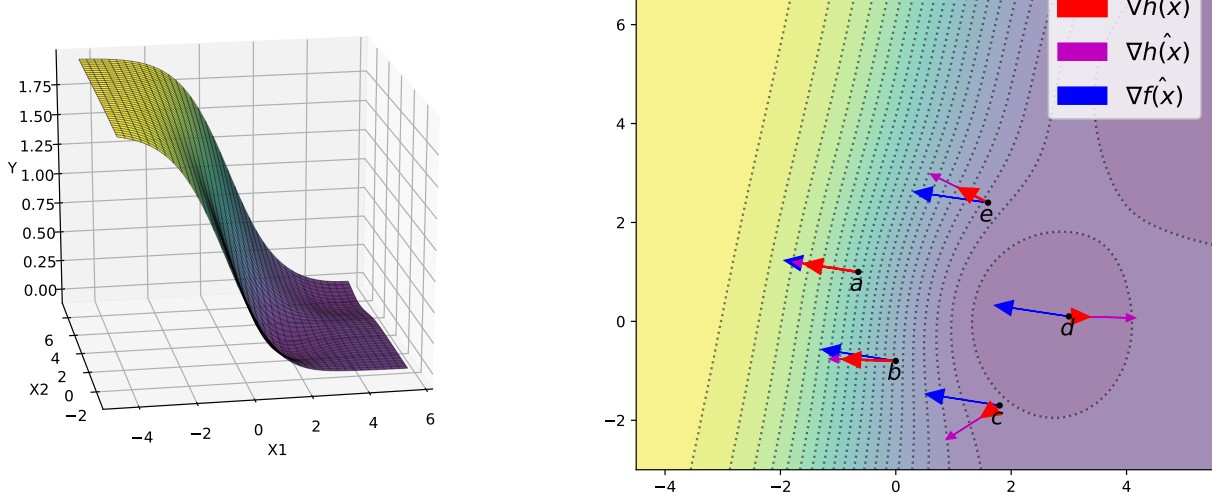

Figure 16: Modified invex function 4, not an invex function.

**Proof:** Let us consider a cone as an initial invex function in Proposition 1. Let us take the cone of form $f(\mathbf{x}) = a\|\mathbf{x} - \mathbf{x}^*\|$, where $\mathbf{x}^*$ is the center/tip of the cone. This is a cone with a scale factor of $a$. The unit vector of the gradient of the cone is given by the following equation.

$$\hat{\nabla} f = \frac{\nabla f(\mathbf{x})}{\|\nabla f(\mathbf{x})\|} = \frac{\mathbf{x} - \mathbf{x}^*}{\|\mathbf{x} - \mathbf{x}^*\|} \tag{10}$$

Moreover, let us consider the cone to be a generalized function. When $a \to 0$ then $f(\mathbf{x}) \to 0$ but $\hat{\nabla} f$ remains the same. The magnitude of this generalized function at all the points is zero, but the direction of the gradient points away from the minima (tip of the cone). Using this in Proposition 1, we get.

$$\nabla g(\mathbf{x}) \cdot \frac{\mathbf{x} - \mathbf{x}^*}{\|\mathbf{x} - \mathbf{x}^*\|} > 0 \tag{11}$$

The method of constructing invex function as mentioned above can not construct all invex functions. Hence, there is a need for a universal invex function constructor.

**Proposition 3** *Modifying invex function as shown in **Proposition 1** for N iterations with non-linear $g(\mathbf{x})$ can approximate any invex function.*

**Intuitive Visual Proof**: The invex function according to Proposition 2 is very simple and can not approximate all invex functions. Whereas, Proposition 1 requires an invex function to start from and modify the function. We can build a basic invex function using Proposition 2 and modify it using Proposition 1 for multiple iterations to make it more and more complex function to approximate the required invex function. A simple intuitive visualization of the requirement of multiple iterations of modification as well as the universality is shown in Figure 17.

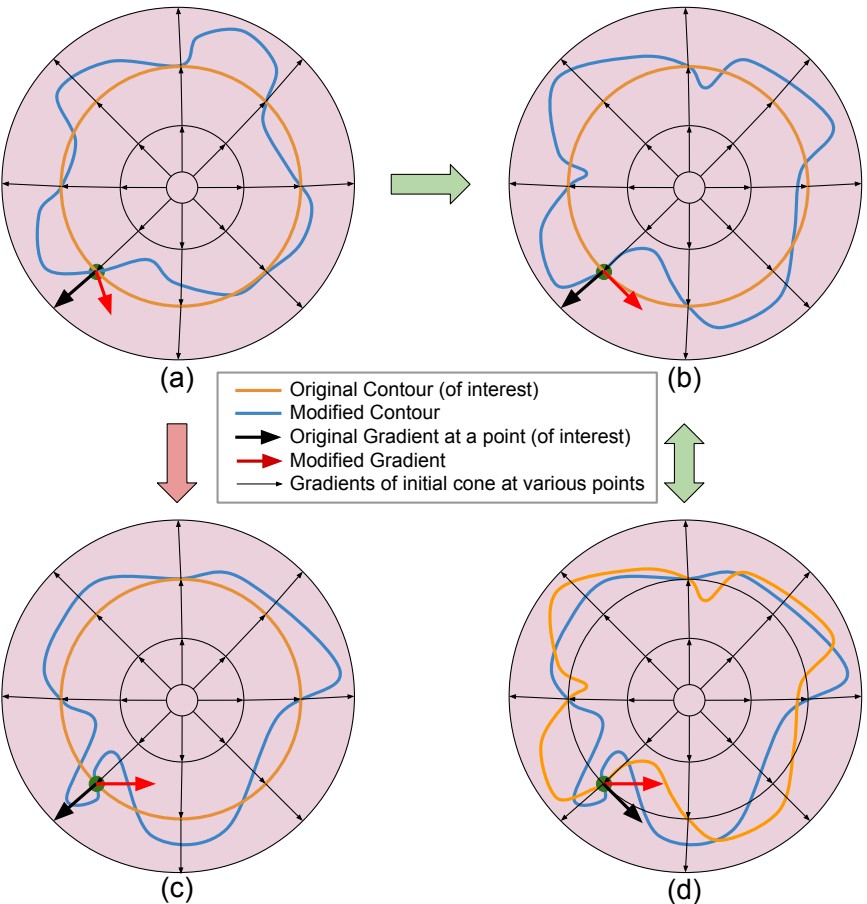

Figure 17: Multiple possible modifications of invex function (or a cone function). The diagrams are made simple by focusing only on a single contour (a 1-connected decision boundary) as well as on a single point of the function. **(a)** Modification of cone showing change in direction $< \frac{\pi}{2}$ or dot product $> 0$ *(as per Proposition 1, 2)*. **(b)** Modification of cone showing change in direction $\simeq \frac{\pi}{2}$. **(c)** Modification of cone showing a change in direction $> \frac{\pi}{2}$ or dot product $< 0$ which is still a connected set; however, our method does not allow such change; hence the requirement of multiple iterations of modification. **(d)** Modification of cone similar to (c), but using twice modified function (a - b - d) showing change in dot product $> 0$ in each step but overall change $< 0$.

In a practical case, let us consider the invex function for the classification-1 dataset. We have done experiments on Table 2 for connected sets using composed invex function. The visualization for the decision boundary is shown in Figure 18.

The initial invex function is guided by the radial gradients as shown in Figure 17. This invex function is limited by Proposition 2. It can't have gradients that oppose the guiding gradients. Hence, we compose on top of this function for a more non-linear invex function as shown in Figure 17. This time the gradient

guidance is enough for 100% accurate classification. Furthermore, this function can be made more non-linear by composing it multiple times. Hence, we can say that multiple iterations of modifying the invex function can approximate any invex function.

## C.3 Algorithm for constructing II-NN

### C.3.1 Modifying II-NN

A Modified II-NN is constructed according to Proposition 1. The parameters of the existing invex function are frozen and the function is modified. The modification is made by adding a new Neural Network to the existing invex function or II-NN. Its corresponding pseudocode is in Algorithm 3.

---

**Algorithm 3:** Modified II-NN - PyTorch like pseudocode

```
# X, t is the dataset with m elements.
# g_iinn is a model with neural network parameter and center parameter.
# f_invex is an invex function (maybe a Neural Network) that is not trained.
# lamda is the scaling parameter for projected gradient penalty.
# f_pg_scale and f_out_clip are gradient penalty and
#   output_gradient clipper respectively as shown in Figure 4
for step in range(STEPS):
    ## 1. forward pass
    y = g_iinn(X) + f_invex(X)
    ## 2. Gradient of the function
    g_gradX, f_gradX = torch.autograd.grad(y, X)
    ## 3. Projected gradient
    pg = torch.bmm(g_gradX.reshape(m, 1, -1),
                    f_gradX.reshape(m, -1, 1)).reshape(-1,1)
    ## 4. Compute projected gradient penalty
    pgp = f_smoothl1(f_pg_scale(pg)).mean() * lamda
    ## 5. Compute gradient of parameters
    pgp.backward_to(g_iinn.parameters())
    ## 6. Compute del_y from loss function
    del_y = criterion(y, t).backward_to(y)
    ## 7. clip del_y using projected gradient
    clip_val = f_out_clip(pg)
    del_y_clip = del_y.clip(-clip_val, clip_val)
    ## 8. Compute gradient of parameters from above
    del_y_clip.backward_to(g_iinn.parameters())
    ## 9. Update the parameters
    optimizer.step()
```

---

### C.3.2 II-NN by guiding with invex function

An II-NN can also be trained by scaling the output of the existing invex function to zero. This is similar to Proposition 2 but instead of a generalized cone, it uses II-NN or other invex function. Its corresponding pseudocode is in Algorithm 4.

## C.4 II-NN Details : Experiment

The Regression 1 dataset used is the same as in Figure 9. The Classification 1 dataset along with predictions by Linear, Convex, Ordinary, Basic Invex and Composed Invex Neural Networks is in Figure 18. We conduct experiments per class basis on MNIST using MLP in Table 6 and CNN in Table 7 as well as on F-MNIST using CNN on Table 8. The experiments compare the performance of the Convex Classifier, Invex Classifier

**Algorithm 4:** Guided II-NN - PyTorch like pseudocode

```python
# X, t is the dataset with m elements.
# g_iinn is a model with neural network parameter and center parameter.
# f_invex is an invex function (maybe a Neural Network) that guides the
#   projected gradient of g_iinn
# lamda is the scaling parameter for projected gradient penalty.
# f_pg_scale and f_out_clip are gradient penalty and
#   output_gradient clipper respectively as shown in Figure 4
for step in range(STEPS):
    ## 1. forward pass
    y = g_iinn(X)
    z = f_invex(X)
    ## 2. Gradient of the function
    g_gradX = torch.autograd.grad(y, X)
    f_gradX = torch.autograd.grad(z, X)
    ## 3. Projected gradient
    pg = torch.bmm(g_gradX.reshape(m, 1, -1),
                   f_gradX.reshape(m, -1, 1)).reshape(-1,1)
    ## 4. Compute projected gradient penalty
    pgp = f_smoothl1(f_pg_scale(pg)).mean() * lamda
    ## 5. Compute gradient of parameters
    pgp.backward_to(g_iinn.parameters())
    ## 6. Compute del_y from loss function
    del_y = criterion(y, t).backward_to(y)
    ## 7. clip del_y using projected gradient
    clip_val = f_out_clip(pg)
    del_y_clip = del_y.clip(-clip_val, clip_val)
    ## 8. Compute gradient of parameters from above
    del_y_clip.backward_to(g_iinn.parameters())
    ## 9. Update the parameters
    optimizer.step()
```

| Class | Architecture | | | | | Invexity % | |
|---|---|---|---|---|---|---|---|
| | Logistic | Convex | Invex (Basic) | Invex (Invertible) | Ordinary | Train+Test | 1M random + (Train + Test) |
| 0 | 98.724 | 99.490 | 99.541 | 99.592 | 99.541 | 100.0000 | 100.0000 |
| 1 | 98.590 | 99.427 | 99.471 | 99.604 | 99.604 | 99.9971 | 99.9998 |
| 2 | 95.155 | 98.401 | 99.079 | 98.789 | 98.983 | 100.0000 | 89.5211 |
| 3 | 94.802 | 98.366 | 98.861 | 98.762 | 98.861 | 99.9843 | 73.8646 |
| 4 | 96.894 | 98.727 | 99.032 | 99.185 | 99.236 | 99.9971 | 99.9998 |
| 5 | 93.330 | 98.430 | 98.711 | 98.99 | 98.599 | 100.0000 | 100.0000 |
| 6 | 97.338 | 99.165 | 99.322 | 99.374 | 99.322 | 99.9986 | 97.2588 |
| 7 | 96.449 | 98.492 | 98.687 | 98.346 | 98.589 | 100.0000 | 33.2285 |
| 8 | 91.273 | 96.150 | 98.665 | 98.409 | 98.871 | 99.9986 | 99.9999 |
| 9 | 93.162 | 96.283 | 98.167 | 98.365 | 98.464 | 99.9871 | 99.9985 |
| Argmax | 90.790 | 96.830 | 97.090 | 97.250 | 97.670 | - | - |

Table 6: MNIST MLP: Accuracy with various architectures and Invexity %

and Ordinary Classifier for all experiments. We compare with Logistic regression on the MNIST dataset. We also test the percentage of the Invexiy rule followed by all train and test points as well as 1 Million random points. It is observed that most of the classifiers follow our Invex rule on $> 99\%$ of training and test points and random points.

We experiment on toy datasets (see Figure 9) to test the capacity of the Invex Neural Network. First, we compare on regression and secondly on classification dataset. We use a 2D spiral dataset for classification which has binary connected sets as classes and is suitable to test the invex function. On a bigger dataset, we experiment on MNIST using MLP as well as CNN architecture and on Fashion-MNIST (Xiao et al., 2017) dataset on CNN architecture. We compare the performance of the Invex function with Linear, Convex and Ordinary Neural Network on Table 2.

The models for toy datasets are trained for 5000 epochs with a full batch whereas the models for MNIST and F-MNIST are trained for 20 epochs with a batch size of 50. For toy regression and classification we use network with configuration $(2,10,10,1)$ and $(2,100,100,1)$ respectively. And MLP on MNIST, our configuration is $(784, 200, 100, 1)$. In these configurations, the first and the last number represents the input and output dimension, and the rest represents the dimension of hidden layers. For CNN on MNIST and F-MNIST we use configurations (1C, 16C, 32C, GAP, 1) and (1C, 32C, 32C, GAP, 1), where configurations are as: input channel, hidden channels, Global Average Pooling (GAP) and output unit. We train the binary classification model for each class of MNIST and F-MNIST and apply argmax over the probability of each class for multi-class classification. The per-class datasets are balanced during sampling. All convolutional layers have kernel:5, stride:2 and padding:1. For regression we use ELU activation and LeakyRelu for rest. Furthermore, we also compare the performance of the Binary Invex Classifier using the invertible method for the invex function. This architecture is not directly comparable as it uses iResNet architecture. However, we use close matching architecture to compare the performance (available on code at github). We find that both of our models perform relatively similar. Furthermore, we can easily use invertible mapping to map output centroid to input space for visualization.

On the MNIST dataset, since the data are not evenly spaced, we use mixup to increase the diversity of points sampled and penalize functions where the constraint is not followed. On the Classification dataset, we use random input points to check and penalize the function where the constraint is not followed. This helps to constrain the function to be invex in the region where data points are unavailable. However, this does not improve performance much when used on CNN. Hence, in the FMNIST experiment, we do not use the mixup method. *Furthermore, we use mixup on invertible invex method experiments, not meant for direct comparision.*

| Class | Architecture | | | | | Invexity % | |
|---|---|---|---|---|---|---|---|
| | Logistic | Convex | Invex (Basic) | Invex (Invertible) | Ordinary | Train+Test | 1M random + (Train + Test) |
| 0 | 98.724 | 97.755 | 99.388 | 99.949 | 99.235 | 100.0000 | 100.0000 |
| 1 | 98.590 | 99.163 | 99.559 | 99.780 | 99.471 | 100.0000 | 96.5372 |
| 2 | 95.155 | 97.335 | 98.837 | 99.758 | 98.643 | 100.0000 | 100.0000 |
| 3 | 94.802 | 97.574 | 98.713 | 99.653 | 99.257 | 100.0000 | 99.9951 |
| 4 | 96.894 | 97.301 | 99.695 | 99.745 | 99.796 | 99.9857 | 10.0613 |
| 5 | 93.330 | 96.973 | 98.823 | 99.664 | 98.935 | 100.0000 | 100.0000 |
| 6 | 97.338 | 97.390 | 99.530 | 99.687 | 99.478 | 100.0000 | 99.9995 |
| 7 | 96.449 | 96.012 | 99.124 | 99.708 | 99.319 | 100.0000 | 99.9503 |
| 8 | 91.273 | 96.920 | 98.614 | 99.538 | 98.922 | 100.0000 | 6.5421 |
| 9 | 93.162 | 92.815 | 97.869 | 99.257 | 98.117 | 100.0000 | 100.0000 |
| Argmax | 90.790 | 94.680 | 97.850 | 98.820 | 98.080 | - | - |

Table 7: MNIST CNN: Accuracy with various architectures and Invexity %

| Class | Architecture | | | | Invexity % | |
|---|---|---|---|---|---|---|
| | Convex | Invex (Basic) | Invex (Invetible) | Ordinary | Train+Test | 1M random + (Train + Test) |
| 0 | 92.10 | 95.25 | 95.40 | 95.30 | 100.0000 | 99.9739 |
| 1 | 97.75 | 99.10 | 99.55 | 99.10 | 100.0000 | 99.9064 |
| 2 | 90.10 | 93.75 | 94.75 | 94.0 | 100.0000 | 98.4567 |
| 3 | 92.60 | 95.60 | 96.95 | 96.20 | 99.9971 | 99.9158 |
| 4 | 89.05 | 93.65 | 94.50 | 93.85 | 99.9929 | 99.8463 |
| 5 | 98.35 | 99.15 | 99.60 | 98.95 | 100.0000 | 100.0000 |
| 6 | 81.65 | 88.00 | 89.40 | 87.95 | 99.9986 | 99.8244 |
| 7 | 93.75 | 98.50 | 98.85 | 98.80 | 100.0000 | 100.0000 |
| 8 | 97.75 | 98.70 | 99.25 | 98.60 | 99.9971 | 99.9998 |
| 9 | 95.90 | 98.60 | 98.95 | 98.30 | 99.9986 | 99.9999 |
| Argmax | 81.27 | 87.76 | 89.80 | 87.80 | - | - |

Table 8: F-MNIST CNN: Accuracy with various architectures and Invexity %

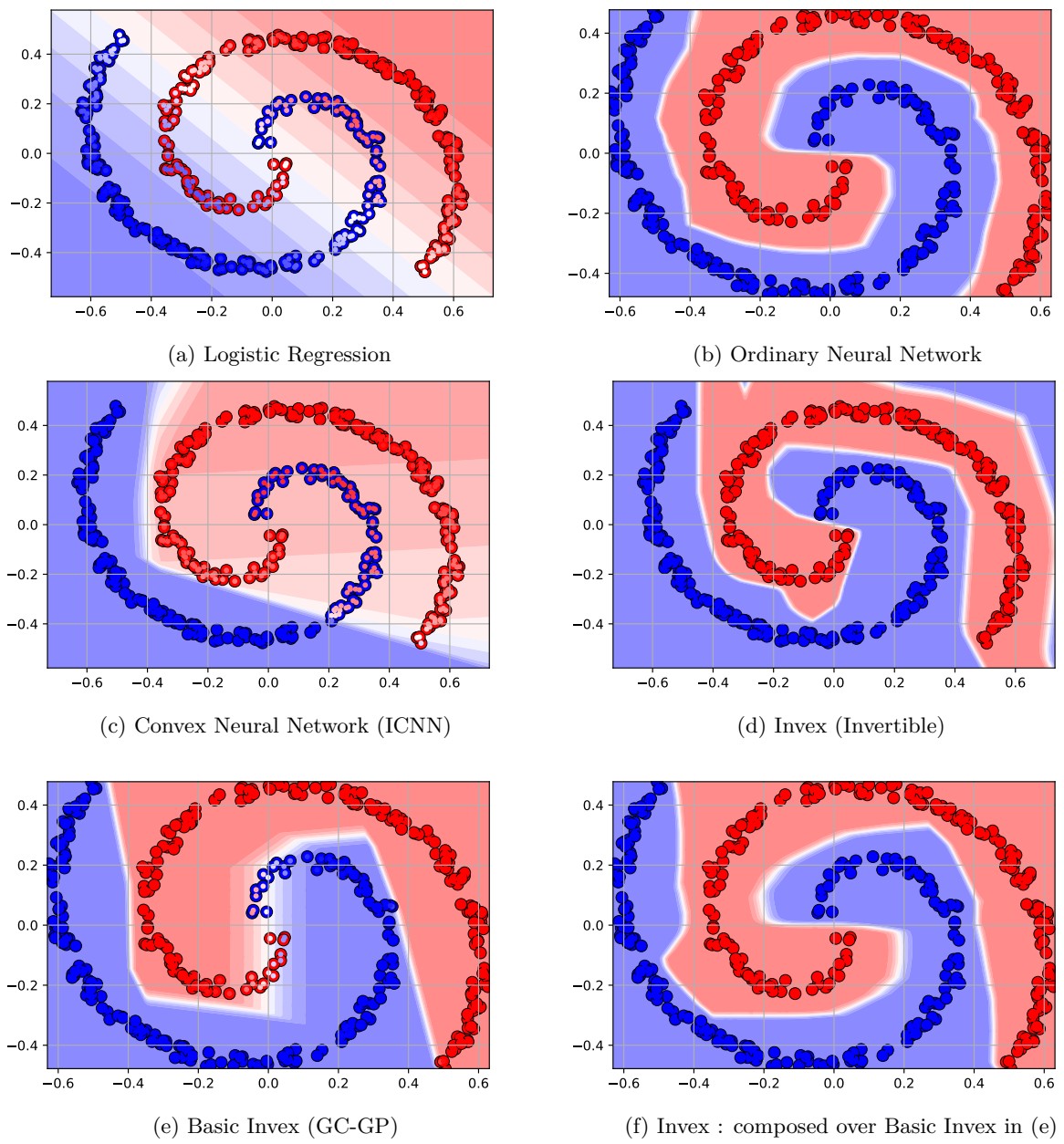

Figure 18: The 2D plot of decision boundary made by (a): Logistic Regression, (b): Ordinary Neural Network, (c): Input Convex Neural Network, (d): Invex (Invertible) Neural Network, (e): Basic Invex and (f): Invex composed over Basic Invex in (e). This dataset is named the Classification 1 task in the tables.

# D  Multiple Invex Classifier

## D.1  Experiment Details

We use an Invertible Residual Network with Convolutional Layers for the Invertible backbone. The classification is performed by connected set classifiers which perform classification by dividing input space into multiple 1-connected sets. We use a convex (1-connected) set using a Voronoi diagram (set) created by linear softmax or distance softmax. We use linear softmax in experiments due to their easier optimization. This

Voronoi set is actually morphed to input space by the invertible mapping to create multiple 1-connected sets (not convex).

For MNIST and Fashion MNIST datasets, we use an Invertible Residual Network with the same configuration (Architecture 1). Similarly, for CIFAR-10 and CIFAR-100 datasets, we also use the Invertible Residual Network with the same configuration (Architecture 2) as shown by Algorithm 5.

---

**Algorithm 5:** Configuration of Invertible Backbone for MNIST and CIFAR.

---

```
### FOR MNIST AND FASHION MNIST: FOLLOWING INVERTIBLE ARCHITECTURE CONFIGURATION
## The Input Image Channels is 1 and the total number of dimensions is 784.
## The Redisual Flow Layer takes number of hidden channels as second parameter.
actf = irf.Swish
Invertible_Sequence = [
    nn.BatchNorm2d(1),
    irf.ConvResidualFlow(1, [16], activation=actf),
    irf.InvertiblePooling(2),
    nn.BatchNorm2d(4),
    irf.ConvResidualFlow(4, [64], activation=actf),
    irf.InvertiblePooling(2),
    nn.BatchNorm2d(16),
    irf.ConvResidualFlow(16, [64, 64], activation=actf),
    nn.BatchNorm2d(16),
    irf.Flatten(img_size=(16, 7, 7)),
    nn.BatchNorm1d(16*7*7),
        ]

### FOR CIFAR-10 AND CIFAR-100: FOLLOWING INVERTIBLE ARCHITECTURE CONFIGURATION
## The Input Image Channels is 3 and the total number of dimensions is 3072.
## The Redisual Flow Layer takes number of hidden channels as second parameter.
actf = irf.Swish
Invertible_Sequence = [
    nn.BatchNorm2d(3),
    irf.ConvResidualFlow(3, [32, 32], kernels=5, activation=actf),
    irf.InvertiblePooling(2),
    nn.BatchNorm2d(12),
    irf.ConvResidualFlow(12, [64, 64], kernels=5, activation=actf),
    nn.BatchNorm2d(12),
    irf.ConvResidualFlow(12, [64, 64], kernels=5, activation=actf),
    irf.InvertiblePooling(2),
    nn.BatchNorm2d(48),
    irf.ConvResidualFlow(48, [128, 128], kernels=5, activation=actf),
    nn.BatchNorm2d(48),
    irf.ConvResidualFlow(48, [128, 128], kernels=5, activation=actf),
    irf.InvertiblePooling(2),
    nn.BatchNorm2d(192),
    irf.ConvResidualFlow(192, [256, 256], kernels=5, activation=actf),
    nn.BatchNorm2d(192),
    irf.ConvResidualFlow(192, [256, 256], kernels=5, activation=actf),
    nn.BatchNorm2d(192),
    irf.Flatten(img_size=(192, 4, 4)),
    nn.BatchNorm1d(3072),
        ]
```

---

Table 9: Accuracy on various Datasets. The blue numbers represent Accuracy using hard classification as per nearest region/classifier for Connected Classifier.. The seeds are $A = 147$, $B = 258$ and $C = 369$. In MNIST, Fashion MNIST and CIFAR-10 experiments, we use 100 connected sets or hidden units for a connected classifier or MLP classifier respectively. For the CIFAR-100 experiment, we use 500 units. *iNN* represents invertible Neural Network as backbone whereas *NN* represents ordinary Neural Network for backbone. *Connected Classifier* represents *our method* of classification and *MLP Classifier* represents ordinary 2 layer MLP for classification task. The settings using *iNN + Connected Classifier* and *iNN + Linear Classifier* produces connected set based classification. **(*)** Data is subject to change.

| | | Dataset (MLP) | Dataset (CNN) | | | |
|---|---|---|---|---|---|---|
| **Architecture** | Seed | MNIST | MNIST | F-MNIST | C-10 | C-100 |
| iNN + Connected Classifier | A | 96.62 | 98.58 | 89.58 | 83.59* | 52.58 |
| | | 96.59 | 98.51 | 88.49 | 83.17* | 51.85 |
| | B | 96.68 | 98.87 | 89.39 | 83.62* | 51.09 |
| | | 96.77 | 98.78 | 86.9 | 83.38* | 50.7 |
| | C | 96.68 | 98.59 | 89.3 | 84.45 | 51.93 |
| | | 96.76 | 98.48 | 87.89 | 84.23 | 50.98 |
| iNN + MLP Classifier | A | 96.57 | 98.2 | 89.37 | 84.48 | 56.91 |
| | B | 96.63 | 98.23 | 89.61 | 84.34 | 56.23 |
| | C | 96.81 | 98.26 | 89.18 | 84.77 | 56.19 |
| NN + Connected Classifier | A | 97.28 | 99.37 | 90.38 | 85.93 | 50.5 |
| | | 97.24 | 99.36 | 88.1 | 85.79 | 50.13 |
| | B | 97.44 | 99.4 | 89.88 | 85.63 | 49.17 |
| | | 97.38 | 99.34 | 88.16 | 85.67 | 48.68 |
| | C | 97.55 | 99.32 | 89.84 | 85.58 | 50.74 |
| | | 97.54 | 99.25 | 88.35 | 85.50 | 50.39 |
| NN + MLP Classifier | A | 97.19 | 98.62 | 90.33 | 85.9 | 52.73 |
| | A | 97.17 | 98.79 | 90.58 | 86.02 | 52.6 |
| | A | 97.38 | 98.89 | 90.35 | 85.75 | 53.08 |
| iNN + Connected Linear | A | 96.63 | 98.16 | 89.35 | 83.75 | 50.78 |
| | | 96.72 | 98.09 | 88.43 | 83.7 | 50.19 |
| | B | 96.59 | 98.38 | 89.07 | 83.95 | 51.95 |
| | | 96.67 | 98.43 | 88.64 | 83.81 | 51.33 |
| | C | 96.83 | 98.17 | 89.32 | 83.86 | 52.44 |
| | | 96.85 | 98.14 | 88.88 | 83.74 | 51.54 |
| iNN + Linear | A | 96.61 | 97.98 | 88.69 | 84.06 | 56.05 |
| | B | 96.61 | 97.96 | 88.54 | 84.71 | 56.08 |
| | C | 96.48 | 98.24 | 88.97 | 84.36 | 56.05 |

The architecture is available with the code. We use our connected classifier that has N hidden units or N connected sets on feature space and output M class probabilities. We know that without Spectral Normalization, the invertible residual network is not necessarily invertible (or ordinary neural network). Since the architecture is created for Invertible Mapping, the performance of models initialized with Spectral Normalization has better performance. The ordinary Neural Network is initialized with Spectral Normalization and then trained without it. Table 9 shows the detailed accuracy metric of each experiments with 3 seeds.

We find that Invertible Backbone helps in optimization as well as provides a reliable backbone for MLP classifiers as well. Experiments show that Connected Classifiers do not have many disadvantages to MLP classifiers, however, we can be certain that the clusters are connected and hence more interpretable. Furthermore, connected classifiers when classified using nearest centroids, we get an insignificant drop in performance.

# E    Requirements for creating Multiple Connected Sets

The algorithm for the Multiple Invex Classifier is shown in the Algorithm 2. The connected classifiers based classification is our novel approach for classification where every region is connected in the input space. If

we transform the data space by an invertible function and apply multiple connected set classifiers, we get connected classifiers in the data space as well.

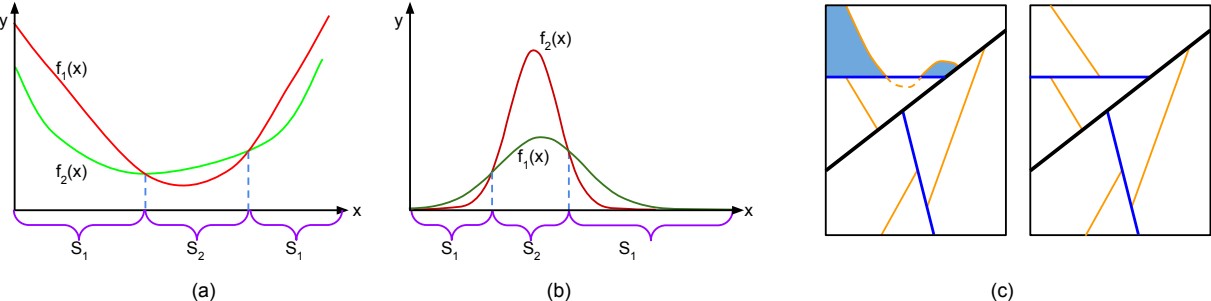

Figure 19: Graphical visualization of type of set formed by various classification methods. (a) Disconnected decision boundary produced by varying convex function. (b) Disconnected decision boundary produced by Gaussian Clusters. (c) Left: Disconnected decision boundary from decision tree due to a non-linear split, right: Connected set due to linear split nodes on decision tree.

### E.1   Argmax over multiple convex/invex functions

Using a simple collection of a convex function, we can find that argmax over different convex functions can produce disconnected sets. Figure 19 shows that argmax over multiple convex sets produces disconnected sets. Hence, we do not use multiple convex functions for multi simply connected set classification.

### E.2   Argmax over Gaussian Mixture Model (GMM)

If the Covariance ($\Sigma$) is not Identity($I$), then GMM may not produce simply-connected sets. This is shown in Figure 19, where we show that GMM with some covariance can produce disconnected sets. This effect is more visible when the individual Gaussian are allowed to be scaled. *(For 2D GMM producing disconnected sets, we provide a code to reproduce such case).* The authors of paper (Izmailov et al., 2020) argue that the GMM Flow is interpretable. We believe that the interpretability is due to the connected sets of the Naive-Bayes classifier with Gaussian Distribution of Identity Covariance. Although they experiment for the Normalizing Flows problem, overall similarity suggests that the Gaussian Mixture Model based decision boundaries might not be simply connected and hence not suitable for interpretability.

### E.3   Argmax over metric functions

We find that argmax over metric functions creates connected sets. Although we do not go into details of metric functions, we find that norm-induced metric functions such as $l2$ and $l1$ norm work for creating connected sets, a Voronoi diagram. Similarly, function like dot product also produces connected sets.

### E.4   Linear decision trees

We also know that linear decision trees create simply-connected sets. We can also use a linear decision tree (decision tree with a plane as a decision boundary) in our multiple connected sets experiments. We find that only linear decision boundaries can guarantee simply connectedness. If we consider convex or slightly non-linear decision boundaries as shown in Figure 19, we find that disconnected decision boundaries may be formed in the global view of the data space. Hence, only linear decision trees can guarantee the 1-connectedness of the decision boundaries.

## F  Invex Function for Optimization

Convex functions are crucial in the field of optimization. The question remains how the invex function can be useful for optimization. Furthermore, gradients of the invex function are also not useful for invertible transformations as the gradients are not monotonous as compared to Convex Potential Flows (Huang et al., 2020).

Using the 2D example as shown in Figure 20, we find that the Invex function does not provide an advantage over the convex function when finding minima. We try Partial Input Invex Neural Network (PIINN) and Partial Input Convex Neural Network (PICNN) for 1D variable optimization. We find that the convex function is sufficient for finding such optimums. Experiments on the 2D toy dataset with 1D variable optimization using PICNN and PIINN get MSE of 0.008806 and 0.0081087 respectively (on a single experiment) which is almost zero. Hence, both of these methods provide a solution to finding minima given partial inputs. An invex function might have different dynamics of reaching the minima, however, we do not experiment for such cases.

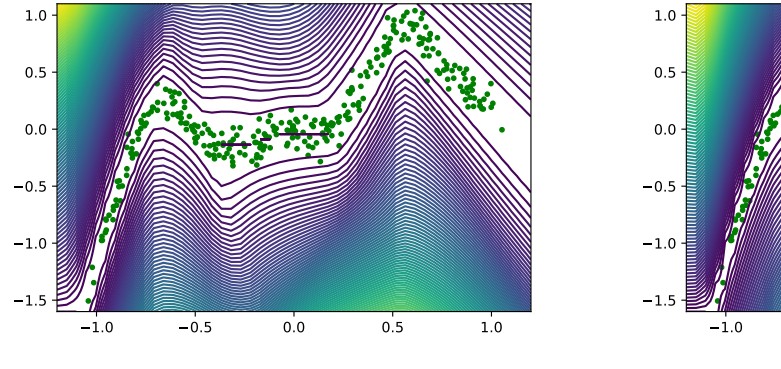

(a) Partial Input Convex Neural Network          (b) Partial Input Invex Neural Network

Figure 20: The plots contain contour plots of partial variable optimization. Here, the x-axis is known and the y-axis is to be optimized. Plot a) and b) show the contour plot of normalized energy function ($E_n = E - E_{min}$).

## G  Multi-Invex classifier on 2D manifold

Manifold can be created by composing Invertible and Linear ($N \rightarrow 2$) functions (Brehmer & Cranmer, 2020). Furthermore, an invex function can be constructed using rank 2 Jacobian transformation, which is equivalent to a manifold, and apply a 2D 1-connected set classifier over it. We use a simply-connected set based multiclass classifier on 2D for the classification of FMNIST and CIFAR-10 datasets. This produces simply connected decision boundary over input space as well as 2D embeddings. Figure 21 shows the manifold embedding on 2D, where the classifier is applied along with the classification decision boundary. We find that 2D embedding-based classification also classifies input data efficiently in FMNIST and CIFAR-10 datasets, however in CIFAR-100, we need a higher embedding dimension for proper classification performance, hence cannot be visualized.

Many methods such as TSNE (Van der Maaten & Hinton, 2008) and UMAP (McInnes et al., 2018) attempt to map input data points to 2D/3D space without using labels which makes comparison impossible.

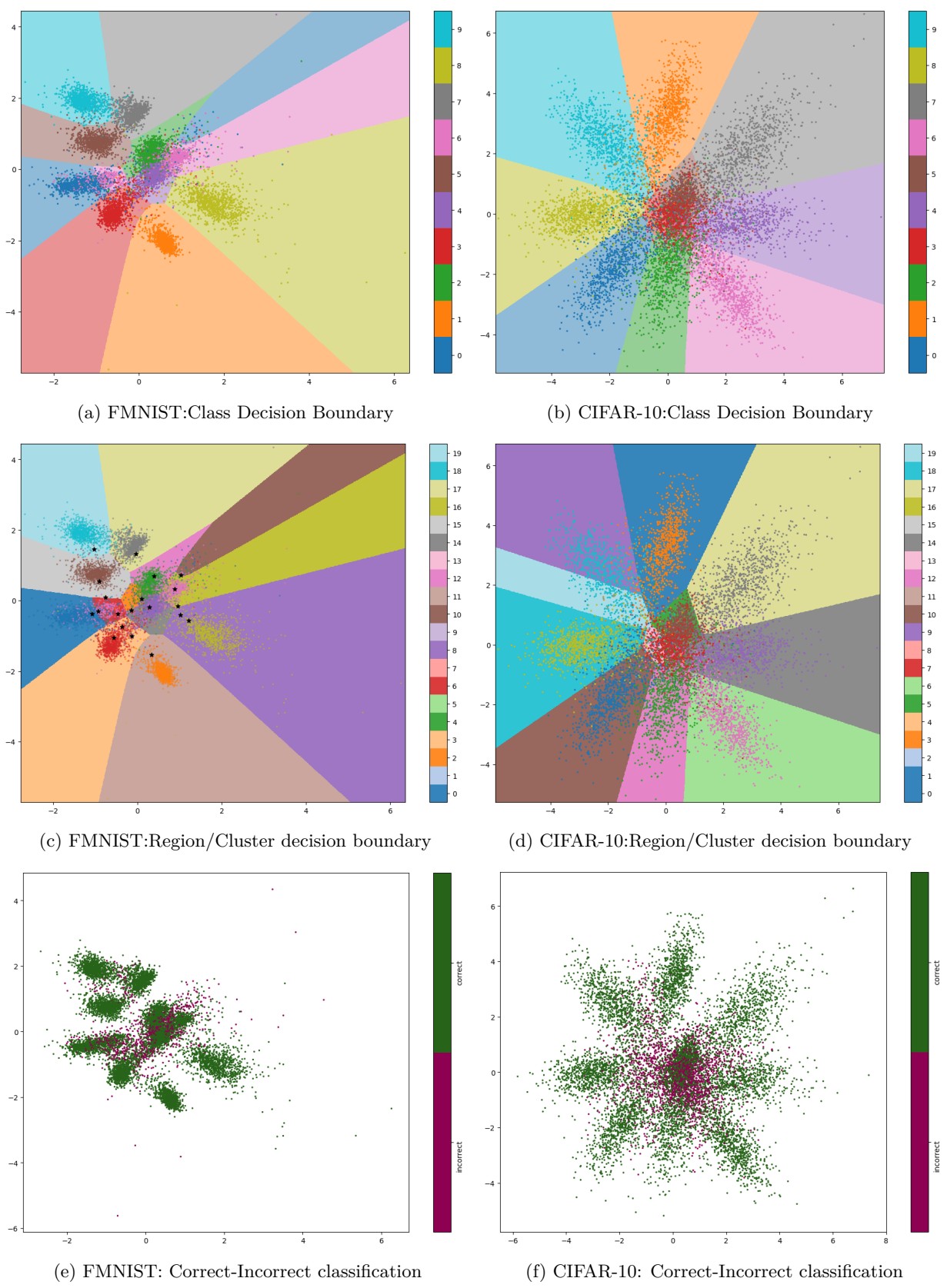

(a) FMNIST:Class Decision Boundary

(b) CIFAR-10:Class Decision Boundary

(c) FMNIST:Region/Cluster decision boundary

(d) CIFAR-10:Region/Cluster decision boundary

(e) FMNIST: Correct-Incorrect classification

(f) CIFAR-10: Correct-Incorrect classification

Figure 21: 2D manifold created by invertible transform and linear projection. Top row *(a and b)* show Class Decision Boundary whereas middle row *(c and d)* show Local Regions for classification. Bottom row *(e and f)* show correctly vs incorrectly classified data points.

## H  Local Classifiers

We use a local classifier, i.e. invex function classifier (bounded 1-connected set) for classification tasks as well as to give a local region around data points. The model works by using multi-invex classifiers with continuous locality scores for classification. An ordinary neural network trained for classification could produce decision boundaries even where data points do not lie. However, if we use local models as shown in the figure 22, we find the decreasing value of local models as we move away from the centroids. This helps us define a local region of influence outside which the model does not predict with confidence, i.e defines that the input is not local to training points. We use tricks like custom gradient function for classification and training using Mean Squared Error. The algorithm is under development and unstable, however, it provides a clue to use bounded 1-connected classifiers to create local regions of influence to detect out-of-distribution data.

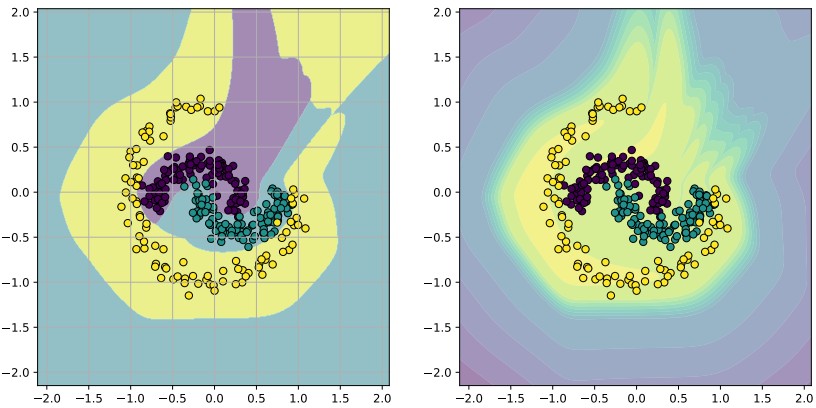

Figure 22: (Left) Decision boundary of the classifier on input space for 3 class 2D-toy-dataset. (Right) Confidence values of local classifiers on the input space.

## I  Invex Function, Manifolds and Poincairé Conjecture

A simplified statement of Poincairé Conjecture is: *every simply connected, closed 3-manifold is homeomorphic to the 3-sphere* (Hosch, 2013). This statement is generalized to n-manifold and n-sphere as: *every topological n-manifold having the same homology and the same fundamental group as an n-dimensional sphere must be homeomorphic to the n-dimensional sphere.*

Let us consider an invertible function $f : Y \in \mathbb{R}^{N+1} \to Z \in \mathbb{R}^{N+1}$ and a convex cone function $g : X \in \mathbb{R}^{N+1} \to \mathbb{R} = \|X\|$. The function $h = g \circ f$ is an invex function according to our Property 1 of invex function.

Let us take the lower contour set of the cone, $g(X) \leq 1$ which is a bounded (or closed) simply connected set (inside of an N-Sphere). If we map this set from $Z$ to $Y$, we get an (N+1)-dimensional closed simply connected set. The mapping can be arbitrary and is defined by the invertible function.

If we take the boundary of the cone, $g(X) = 1$ which is a closed simply connected manifold (an N-Sphere). If we map this manifold from $Z$ to $Y$, we

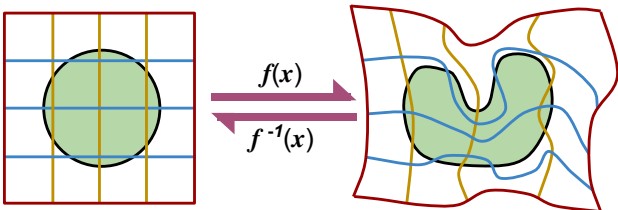

Figure 23: Non-Linear Homeomorphism of Input space; input space has a circular (1-connected) set/manifold; output space has morphed space along with a morphed 1-connected set. The homeomorphism is reversible (and learnable). This property allows n-Sphere to be homeomorphic to every 1-connected (closed) n-Manifold.

get an N-Dimensional closed simply connected man-
ifold on N+1 Dimensional space. The transformation/mapping is done to the whole space of $Z$ and $Y$, however, if we are only concerned with the manifold, we can get arbitrary homeomorphism as defined by the invertible function. This is depicted in Figure 23 for 2-D input space.

We use a similar approach of thresholding the invex function to define if points lie inside or outside simply connected manifold. This allows us to create a simply connected decision boundary for the input space. The decision boundary itself is a simply connected manifold. A similar idea related to Manifolds using invertible function has already been studied (Brehmer & Cranmer, 2020), where the authors reduce the dimension of the invertible backbone using linear projection, which creates a manifold in input space.

## J    Future Works

Although Local Decision boundaries have multiple applications, we only use them for connected set based classification in our paper. We believe that our work can be extended to other areas of Machine learning such as Rejection based classification and rejecting adversarial examples. Furthermore, we believe that our theoretical work can be extended to normalizing flows and towards the interpretability of data space using connected sets. We also believe that clustering data points in input space can help us divide the input space for local learning tasks in Neural Networks . Although we do not find an application of the invex function in general tasks that convex function has found its application, we believe that the invex function has potential application in optimization.

We aim to use the invex function and the concept of the locality to understand/interpret already existing locality in data embeddings (eg. word embeddings in NLP) (Mikolov et al., 2013), and local functions such as Linear-Softmax, SoLU (Elhage, 2022), Attention and Energy based models, Mixture of Experts (MoE) (Masoudnia & Ebrahimpour, 2014; Shazeer et al., 2017) and for the interpretation of input space in general.

