# OpenReview forum: "Input Invex Neural Network"
_TMLR — Rejected by TMLR_

### Review · Reviewer_6TC1 · 2022-10-06

**Summary Of Contributions:**

Broadly, the paper considers the problem of building a neural network which is "invex" wrt its input. Invexity generalizes convexity, in the sense that the NN will separate the space into two generic connected components (instead of a convex and a non-convex component, like for a convex model).

The authors build invex NNs with two techniques:
1. By constraining their gradient in order to satisfy proposition 1 in the paper.
2. By using an invertible ResNet, which is invex by default (property 2 and end of page 9 in the paper).

Because invex functions are scalar valued, they perform multi-class classification by one-vs-all methods (section 3.2.1). They also build a variant with multiple connected components which are linearly combined (section 3.2.2). There are a few experiments that show how invex NNs do not loose too much accuracy compared to standard neural networks.

**Broader Impact Concerns:**

I do not see particular concerns for this.

**Requested Changes:**

- Please rewrite Section 1 to clarify possible applications of invex NNs. It would be helpful to also have at least one worked out example in the experiments apart from standard classification experiments in terms of accuracy.

- Clarify with definitions what is meant in Section 2.1 and how it connects to the rest of the paper.

- Clarify the two algorithms by moving all the relevant bits from the appendices to the paper. I would also suggest to clean as much as possible the pseudocodes (e.g., the torch.randn from above, but there are multiple completely unneccessary instructions).

- Please also remove very informal sentences like "With much experiments, background research and theorizing, we found that such functions are called the invex function".

- "There has been a criticism of the unclear definition of invexity and its related topics such as pre-invex, quasi-invex functions and sets". I have found one paper (https://arxiv.org/abs/1203.1152) with some valid criticisms in this sense, e.g., invexity is so broad that the resulting sets can be of any form and shape. Can the authors spend more time discussing possible limitations along this line?

- "x* be its minima" in the theorems: the minima of what?

**Strengths And Weaknesses:**

I have several concerns on the paper, broadly summarized in terms of motivation, relevance of the results, and most importantly organization and clarity of the paper.

MOTIVATION: first, why do we need invex NNs? The authors state that "ICNN Amos et al. (2017) has been a goto method for constructing convex functions using neural networks." However, ICNN is built to be convex to a subset of its inputs, in order to transform inference (wrt the convex inputs) into an efficient optimization problem.

Maybe I am mistaken, but the only interesting result they have at the moment is a visualization of the centroids of each component (Fig. 7), but is a centroid even a valid measure for a possibly non-convex set? Apart from this, their networks are harder to train and with lower accuracy. I don't think the paper makes a good job at providing easy-to-follow motivations.

There are loose ideas in the paper ("Local neuron modelling with invex functions could be useful for Network Morphism based Neural Architecture Search", "locally activating neurons in Neural Networks", "the interpretability of data space using connected sets"), but they are relegated to future works and in general (at least to me) it is not immediately clear how these invex NNs can be applied in these contexts.

RELEVANCE: of the two techniques they are presenting, gradient clipping is shown as a baseline, while the other method is simply a iResNet followed by a sigmoid. So, the results here are akin to saying that an iResNet maps its embeddings on a connected set. Isn't this trivial from the inverse function mapping?

CLARITY: I'll provide some comments in order wrt the paper.

Section 2.1: "The locality is another idea related to simply connected space". This mention of locality is unclear, if I understand correctly the authors are arguing that one can define locality as "all the points in the same connected component"? However, this idea is never really used in the paper itself.

Section 2.6 ("Invex Function, Manifolds and Poincairé Conjecture"): this is really hard to follow and, again, I am not sure it contributes to the paper itself?

Section 3.1.1: the description of the method is quite hard to follow and the figure (5) does not help. Appendix A.3 is even worse because there are random constants appearing (e.g., "3*pg − 0.0844560006").

Section 3.2.2 ("Multiple Connected Sets"): the paper does not explain how the different components are connected (Fig. 6 only shows each component). The pseudocode in the appendix is worse, there are some combining weights which are initialized randomly (torch.randn), then set to "5" (why 5?). There's a misterious "hard" variant where the authors are multiplying the logits by 1e5.

There are many additional items that would require attention, but these are the most important ones.

---

> ### Author Response · Authors · 2022-11-28
> **Incorporation of Requested Changes and Comments on Weakness**
>
> Thank you for your constructive feedback.  We revised our paper including your feedback. Please see our responses below.
>
> ## Concerns:
>
> **Motivation:**
>
> *- why do we need invex NNs ?*
>
> **Ans:**  There is a need for highly non-linear and general simply connected sets as well, for which we need an invex function (**Table 1**). We need invex NNs in general for learnable simply connected sets. Such connected sets produce higher classification accuracy than convex sets (**Table 2**). Please revisit the revised introduction for more details on the motivation.
>
> To clarify the benefit of using a 1-connected set over a convex set, we experiment on the toy classification problem in **Section: 4.1** (*Multi-convex set vs multi-invex set classification*)  on the revised version.
>
> *- … (Fig. 7), but is a centroid even a valid measure for a possibly non-convex set? Apart from this, their networks are harder to train and with lower accuracy.*
>
> **Ans:** Centroid is not a measure for a non-convex set, in fact, we can visualize centroid for a convex set as well.
>
> Please note convex neural networks are the closest comparable method to our work in terms of binary classification. **Table 2** shows that our method surpasses the convex neural network.
> It is true that our method lags behind the ordinary neural network. However, our approach provides additional benefits of interpretability and local stability which benefits network morphism. We add new paragraphs in **Section 3** *( Why do we need Multi-Invex Classifiers ? )  & ( Interpretability )* explaining this as well as experiments on toy dataset in **Section 4.1**.
>
> *- There are loose ideas in the paper … are relegated to future works*
>
> **Ans:** We added experiments in **Section 4.1** on Network Morphism with local region adding and removing. We have added the interpretation of classifiers in data space using connected sets in **Table 4**.
>
> **Relevance:**
> *- iResNet followed by a sigmoid. ... Isn't this trivial from the inverse function mapping*
>
> **Ans:** We use *iResNet + Linear Classifier* for binary classification which is intuitive and critical for interpretation. And the linear decision boundary on the output space which is connected is also connected on the input space. Furthermore, *iReset + Linear/Cone + Sigmoid* in its combined form is an *invex* function from **Properties 1 and 2** of the invex function in **Section 2.2**. Looking from the perspective of an invex function allows for manifold-based binary classification as well.
>
> **Clarity**
>
> **2.1 Ans:** This concept of locality is actually used in multiple sections and has been clarified in the revision. It is more clear in visualization in the experiment **Sections 4.1** and  **4.2**.
>
> **2.6 Ans:** We move the section to Appendix. It is relatable to the idea of a 1-connected set or 1-connected manifold. However, the details are not relevant to the applications shown in the experiments.
>
> **3.1.1 Ans:** The figure helps explain the algorithm for GC-GP based invex function. The functions used are developed by graph plotting and the random constants are not meaningful, but just represent the plotted function.
>
> **3.2.2 Ans:** A toy experiment in **Section 4.1** (Multi-convex set vs multi-invex set classification) might better show the connected components. These components are Voronoi diagrams and are connected accordingly.
>
> We clean up the pseudocode. We used torch.randn, which produces (-4 , 4) at max, hence to pre-set each region to a class, we set its value to 5 which serves as an initialization method. However, we remove this for simplifying the algorithm to its core function.
>
> The *hard* variant performs *softmax* with temperature of 1e5, (now 1e9) which is equivalent to performing argmax over the regions, i.e. only 1 neuron in the connected classifier activates for the hard variant as well as is interpretable as a single region. The selected region from argmax has a class probability which is the output of the region.
>
> ## Requested Changes:
>
> *(1)* We have revised *Section 1*. We add network morphism and interpretability as an additional application of our method. See **Section 4.1**  (*Network Morphism: Adding and Removing Regions*) for network morphism and Section 4.2 (*Interpretation of Regions/Neurons*) for experiments on interpretability.
>
> *(2)* We make slight modifications to the definition of locality and its use in the rest of the paper.
>
> *(3)* We clean the pseudocode and move to the main content (**Section 3**)
>
> *(4)* We modify that.
>
> *(5)* We cite the paper, we find that the high dimension connected sets might be overfitting and we need methods to regularize that. We mention that in the limitation **Section 5**. We do not go in depth in the possible resulting sets.
>
> *(6)* We fix that to be minima of $f$ in the equations.

---

### Review · Reviewer_gVf6 · 2022-11-01

**Summary Of Contributions:**

This paper presents methods for constructing connected decision boundaries, called invex function, using neural networks. The first method to construct the invex function uses gradient clipped gradient penalty (GC-GP), and the second method composes the invertible function and convex function. The authors argue that such connected decision boundaries are interpretable. In the experiment, the authors applied the proposed input invex neural networks (II-NN) to simple classification tasks and show that II-NN exhibits comparable performance to ordinary neural networks.

**Broader Impact Concerns:**

The reviewer thinks that there is no ethical concern for this paper.

**Requested Changes:**

[Major Comments]
1. The advantage of II-NN against invertible and ordinary neural networks is unclear and should be clarified theoretically or experimentally. It would be better to include the performance of invertible neural networks for experimental comparison. Also, it would be nice to evaluate the interpretability of II-NN against other methods.

1. The variation of the proposed method should be clarified. For example, what does $\lambda$ mean in Table 2, and how to tune this parameter? what is the difference between "Basic Invex," "Invex (composed)," and "Invex (Invertible)"? What does "iNN" mean in Table 3?

1. The reviewer did not find the definition of "centroid" in this paper. Although the reviewer guesses that the visualizations in the MNIST and Fashion-MNIST are given by using the invertible property of II-NN, the reviewer could not understand the detailed procedure of the visualization. Therefore, the detailed definition and calculation of "centroid" should be clarified.

**Strengths And Weaknesses:**

[Strengths]
* A novel concept, invex function on neural networks, is introduced. The theoretical foundation seems to be sound.
* The experimental result shows that the proposed II-NN exhibits comparable performance to ordinary neural networks in the simple MNIST and Fashion-MNIST datasets.

[Weaknesses]
* Although the authors mentioned that an advantage of II-NN is its interpretability, the experimental evaluation regarding interpretability seems to be weak. The reviewer could not understand the advantage of II-NN against existing convex or invertible NNs in terms of interpretability.
* The invertible neural networks should be the competitor for II-NN. However, the experimental comparison is conducted with only convex and ordinary neural networks.

---

> ### Author Response · Authors · 2022-11-28
> **Incorporation of Requested Changes**
>
> Thank you for your constructive feedback. We revised our paper including your feedback. Please see our responses for the requested changes below.
>
> 1) The advantage of II-NN against invertible and ordinary neural networks is unclear and should be clarified theoretically or experimentally. It would be better to include the performance of invertible neural networks for experimental comparison. Also, it would be nice to evaluate the interpretability of II-NN against other methods.
>
> **Ans:** We have added the advantage of II-NN against ordinary neural networks as well as against invertible+linear classifiers in **Section 3.2.2** (*Why do we need Multi-Invex Classifiers ?*) as well as experiment with it in **Table 3** of **Section 4.2**. You can find the theoretical advantage of IINN over invertible and ordinary networks as compared to convex classifiers in terms of interpretability, generality and performance.
>
> 2) The variation of the proposed method should be clarified. For example, what does λ mean in Table 2, and how to tune this parameter? what is the difference between "Basic Invex," "Invex (composed)," and "Invex (Invertible)"? What does "iNN" mean in Table 3?
>
> **Ans:** We have made clear the role of lambda and how it was tuned with a short discussion (**Section 4.2**). However, we do not conduct rigorous experiments tuning the hyperparameters. Tuning rigorous hyper-parameters may add accuracy to our method. We also clarify the terms  *"Basic Invex," "Invex (composed)," and "Invex (Invertible)"* in **Table 2**  and *"iNN"*  in **Table 3**.
>
> 3) The reviewer did not find the definition of "centroid" in this paper. Although the reviewer guesses that the visualizations in the MNIST and Fashion-MNIST are given by using the invertible property of II-NN, the reviewer could not understand the detailed procedure of the visualization. Therefore, the detailed definition and calculation of "centroid" should be clarified.
>
> **Ans:** We have added a simple explanation “that the centers used in euclidean distance/cone based classification can be reversed to input space to visualize the centroid” in **Section 3.1.2** and in experiments (*Interpretation of Regions/Neurons*).

---

### Review · Reviewer_KmDH · 2022-11-03

**Summary Of Contributions:**

Authors develop two ways to construct invex function by neural networks. The first method constrain the gradient by using a new gradient clipping method; the second one combines an invertible and a convex neural network. Authors provide justifications why these two methods can learn invex function. As an application, the current experiment suggests that the developed input invex neural network achieves good results on classification.

**Broader Impact Concerns:**

No ethical concerns.

**Requested Changes:**

1. Please reorganize the structue of this paper. Give more details in Section 3.1.1. If possible, authors can also consider move Section 3.3 to Appendix or future work.
2. Please give more justfication to Section 5 Limitation. Please justify the two arguments more rigorous.

**Strengths And Weaknesses:**

Strengths:
1. The movation to use neural networks to construct invex function and its usages to improve classification interpretability is interesting.
2.  Authors aim to investigate the utility of their developed input invex neural network to various machine learning problems, including classification, regression, and optimization (although some arguments are not well supported).

Weaknesses:
1. I should say one of the biggest weaknesses of this paper is that it is a little difficult to follow. For example, the main part of this manuscript should be Section 3.1. However, many details are missing in both 3.1.1 and 3.1.2. I mean readers need to check the Appendix multiple times to understand output gradient-clipping and projected gradient-penalty. I would suggest move some materials in Appendix (especially the equations) to Section 3.1.1 to improve the readability. Moreover, it would be much more helpful if authors can explicitly mention which proposition that each method relies on. It is very unclear if authors just say "based on these propositions".

2. Some arguments are not well supported in this paper:
2.1. Authors mention in Section 3.3 that II-NN can be used for regression and optimization as well. But almost all experiments are done for classification scenario. Why not move Section 3.3 to Appendix or list as future work? Especially since some properties are still unclear or just tested on a toy data.
2.2. I am not sure how to understand "knowledge of having simply connected decision boundaries is a step towards interpretability" in the Conclusion part. Does the interpretability comes from the visualization of the centroid of the set? On the other hand, how is the results shown in Fig. 7 related to the concept of "prototype" (in explainable AI)?
2.3. Authors mention in the Limitation part that "Invex function using gradient constraint depends on GC-GP, a robust method, however, it can not guarantee if the function is invex or not". Is this a severe issue? On the other hand, how to understand the following two arguments? I feel both arguments are not well supported.

---

> ### Author Response · Authors · 2022-11-28
> **Incorporation of Requested Changes and Comments on Weakness**
>
> Thank you for your constructive feedback. We revised our paper including your feedback. Please see our responses below.
>
> **Concern:**
>
> *1) I would suggest move some materials in Appendix (especially the equations) to Section 3.1.1 to improve the readability. Moreover, it would be much more helpful if authors can explicitly mention which proposition that each method relies on.*
>
> *2.1) … But almost all experiments are done for classification scenario. Why not move Section 3.3 to Appendix or list as future work? …*
>
> **Requested Change:**
>
> *1) Please reorganize the structue of this paper. Give more details in Section 3.1.1. If possible, authors can also consider move Section 3.3 to Appendix or future work.*
>
> **Ans:**  We move equations and the algorithm to the main S**ection 3.1.1**. We also properly mention the exact use of the propositions in the experiments in **Table 2** as well as in **Figure 4** and **Algorithm 1** in **Section 3.1.1**.
> We also removed the regression (*Section 3.3*) part as it did not contribute much. We have moved the invex function for optimization to **Appendix F**.
>
> **Concern:**
>
> *2.2. I am not sure how to understand "knowledge of having simply connected decision boundaries is a step towards interpretability" in the Conclusion part. Does the interpretability comes from the visualization of the centroid of the set? On the other hand, how is the results shown in Fig. 7 related to the concept of "prototype" (in explainable AI)?*
>
> **Ans:**  We have added the experiments and discussion on the “Simply connected sets and Interpretability” in multiple sections. In **Section 3.2.2** (*Why do we need Multi-Invex Classifiers ?*), we discuss the Voronoi Partitions and their relation to our method and its interpretability as well as discuss the relationship with the concept of *Prototype in Explainable AI*. Inspired by this, we have also added the experiments on interpretability (**Section 4.2:** *Interpretation of Regions/Neurons*) for CIFAR-10 with visualization of center, medoid and nearest samples of a region/cluster along with statistics of the region on the test samples.
>
> **Requested Change:**
>
> *2) Please give more justfication to Section 5 Limitation. Please justify the two arguments more rigorous.*
>
> **Ans:** We have also explained the limitation of GC-GP with relation to percentage(%) of points following the invex rule. We also simplify our two arguments (clarify first, remove second) to give a simple idea of what we are trying to convey in **Section 5**.

---

> > ### Comment · Reviewer_KmDH · 2022-12-10
> > **Thanks for the revision**
> >
> > Good to see that authors add more details on the interpretability and also discussed its relationships to the prototype. The manuscript is now much easier to follow, and most of my concerns are addressed.
> >
> > Just a simple question:
> > In the definition in Eq. (3), it should be $\nabla f(x_2)$, rather than $\nabla f(x_1 - x_2)$?

---

> > > ### Author Response · Authors · 2022-12-11
> > > **Correction of equation**
> > >
> > > Thank you for pointing this out. We made the changes on the paper.

---

### Decision · Action_Editors · 2023-01-04

**Recommendation:** Reject

**Comment:**

The paper addresses an issue that can be of potential interest, i.e., in a nutshell, learning avoiding local minima by construction. Most of the reviewers have recognised a positive contribution of the paper, however I agree with one of the reviews that the current version of the paper fails to provide enough support for the use of the proposed models given the lack of a proper motivation, and theoretical results/empirical evidence of superiority with respect to standard neural network models.
In general:
- Presentation: wording and concept definition should be improved, e.g. ”However, we need a method that gives us a convex set that is always simply connected but not necessarily required to be a convex set.” is contradictory since there is the requirement to have a convex set that is not necessarily a convex set! Concerning missing concept definitions, an example is the mention of the “lower contour set” concept without having defined it.
- Interpretability/Explainability: “We design our qualitative experiments to support our claims on the interpretability and explainability of our method compared to our Neural baseline architectures. Please note comparing our method with the work on AI and explainability would be beyond the scope.” If the aim of the paper is not to compare versus existing approaches for interpretability and explainability, then there is no reason to design the experiments to support claims on the interpretability and explainability of the proposed approach. Either this is done and then a comparison versus state-of-the-art approaches is reported to carefully place the proposed models in the current relevant literature, or this is not done, and such assessment study is postponed to future work. If you think (as I personally think) this property is fundamental to make relevant and significative your proposal, then it should be done in a proper way.
- Main contribution: the argument about implementing disconnected regions as the union of connected regions is vacuous since the issue is how many of those disconnected regions we need to correctly implement the target function. The more disconnected regions we need to characterise each single class, the less it is relevant that each single region should be connected. The paper fails in addressing this basic question. Basically what is missing in the current version of the paper is a characterisation of the bias introduced by invex functions, and how such bias is modified when introducing the union of a set of connected regions as representation of a single class. Specifically, what is the difference with respect to the bias introduced by soft decision trees or neural trees (https://iopscience.iop.org/article/10.1088/0954-898X/1/4/003, https://proceedings.mlr.press/v97/tanno19a.html), that follow a similar strategy by building the union of (convex/non-convex) regions to characterise each single class. An experimental comparison versus soft decision trees/neural trees would also be needed.

Authors are encouraged to address all the above issues, clearly explaining in the introduction why TMLR's audience should be interested to use invex neural networks instead of general neural networks, before reconsidering a new submission to TMLR.

**Audience:**

Because of the issues about claims and evidence, the paper, in its current form, does not provide enough material to make individuals in TMLR's audience eagerly interested in the findings of the paper.

**Claims And Evidence:**

The current version of the paper does fail to give a convincing motivation for the usefulness of invex neural networks. Theoretical justification for the superiority of the proposed approach is not provided, and empirical evidence does not support such superiority. Informal arguments about the usefulness of the proposed models are not well explained and convincing. E.g, how different are they from soft decision trees / neural trees ? Improved interpretability and explainability has been raised up as a property of invex neural networks, however a proper assessment versus SOTA approaches in the field is not provided.